# Human pluripotent reprogramming with CRISPR activators

Jere Weltner[1], Diego Balboa [1], Shintaro Katayama[2], Maxim Bespalov[1], Kaarel Krjutškov[2,3], Eeva-Mari Jouhilahti[1], Ras Trokovic[1], Juha Kere [1,2,4,5] & Timo Otonkoski [1,6]

CRISPR-Cas9-based gene activation (CRISPRa) is an attractive tool for cellular reprogramming applications due to its high multiplexing capacity and direct targeting of endogenous loci. Here we present the reprogramming of primary human skin fibroblasts into induced pluripotent stem cells (iPSCs) using CRISPRa, targeting endogenous *OCT4*, *SOX2*, *KLF4*, *MYC*, and *LIN28A* promoters. The low basal reprogramming efficiency can be improved by an order of magnitude by additionally targeting a conserved Alu-motif enriched near genes involved in embryo genome activation (EEA-motif). This effect is mediated in part by more efficient activation of *NANOG* and *REX1*. These data demonstrate that human somatic cells can be reprogrammed into iPSCs using only CRISPRa. Furthermore, the results unravel the involvement of EEA-motif-associated mechanisms in cellular reprogramming.

[1] Research Programs Unit, Molecular Neurology and Biomedicum Stem Cell Centre, Faculty of Medicine, University of Helsinki, Helsinki 00014, Finland. [2] Department of Biosciences and Nutrition, Karolinska Institutet, Huddinge 141 83, Sweden. [3] Competence Centre on Health Technologies, Tartu 50410, Estonia. [4] School of Basic and Medical Biosciences, Guy's Hospital, King's College London, London SE1 9RT, UK. [5] Folkhälsan Institute of Genetics, Helsinki 00290, Finland. [6] Children's Hospital, Helsinki University Central Hospital, University of Helsinki, Helsinki 00290, Finland. These authors contributed equally: Juha Kere, Timo Otonkoski. Correspondence and requests for materials should be addressed to J.W. (email: jere.weltner@helsinki.fi) or to J.K. (email: juha.kere@ki.se) or to T.O. (email: timo.otonkoski@helsinki.fi)

CRISPRa system relies on sequence specific recruitment of a catalytically inactivated version of Cas9 protein (dCas9) to genomic sequences defined by short guide RNA (gRNA) molecules[1-3]. The fact that dCas9 effectors can be used to control transcription of targeted endogenous loci makes it useful for mediating cellular reprogramming, which requires silencing and activation of endogenous gene sets for proper cell type conversion. CRISPRa may therefore be beneficial in overcoming reprogramming barriers that limit reprogramming efficiency and contribute to the emergence of partially reprogrammed stable cell populations, often associated with inadequate endogenous gene activation or silencing[4-6]. Previously, dCas9 effectors have been used to mediate differentiation, transdifferentiation, and reprogramming of various mouse and human cell types, but complete pluripotent reprogramming of human cells using only CRISPRa has not yet been reported[7-15].

In addition to gene activation, dCas9 effector mediated DNA targeting can be used to decipher the functions of genomic regulatory elements[16-18]. Combining reprogramming factor promoter targeting gRNAs with targeting of other regulatory elements has high potential in mediating comprehensive resetting of gene regulatory networks. A conserved Alu-motif was recently reported to be enriched in the promoter areas of the first genes expressed during human embryo genome activation (EGA)[19]. This sequence is thus likely to be involved in the control of early embryonic transcriptional networks. As human embryos can reprogram somatic cell nuclei[20], we hypothesized that targeting this EGA-enriched Alu-motif (EEA-motif) could enhance reprogramming of somatic cells to pluripotency.

Development of reprogramming approaches for faithful recapitulation of cellular phenotypes is an important task, considering the increasing pace with which reprogrammed cells are moving toward clinical trials[21]. Here we describe a method for reprogramming human cells, including primary adult human skin fibroblasts, into induced pluripotent stem cells by CRISPRa. This opens up important possibilities for the development of more extensive CRISPRa reprogramming approaches for human cells. Efficiency of the method depends on the targeting of the EEA-motif, which results in improved activation of a subset of endogenous genes that work as reprogramming factors, including *NANOG* and *ZFP42* (*REX1*). These results also exemplify the potential in targeting cell type enriched regulatory elements for controlling cell fate.

## Results

**CRISPRa-mediated reprogramming of NSCs and EEA targeting.** We began human cell reprogramming with CRISPRa using a simplified reprogramming scheme. CRISPRa-mediated *POU5F1* (*OCT4*) activation has been used to replace transgenic OCT4 in human fibroblast reprogramming, while the transgenic expression of only OCT4 has been shown to be sufficient for the reprogramming of neuroepithelial stem cells (NSCs) into iPSCs[12,22]. We therefore combined CRISPRa-mediated *OCT4* activation and NSC reprogramming as an initial model using trimethoprim (TMP) stabilized SpdCas9VP192 fused with P65-HSF1 activator domain[23] (DDdCas9VPH) under doxycycline (DOX) inducible promoter (Fig. 1a, b). Expression of DDdCas9VPH and *OCT4* targeting guides in iPSC-derived NSCs resulted in the emergence of pluripotent cells in a DOX and TMP dependent manner (Fig. 1c–e). These cells could be expanded into stable dCas9 independent cell lines (Fig. 1c and Supplementary Fig. 1). This demonstrated that CRISPRa mediated activation of endogenous *OCT4* alone was sufficient to reprogram NSCs to iPSCs.

To determine if EEA-motif targeting could improve CRISPRa reprogramming of NSCs, we designed a set of five 14 nt gRNAs

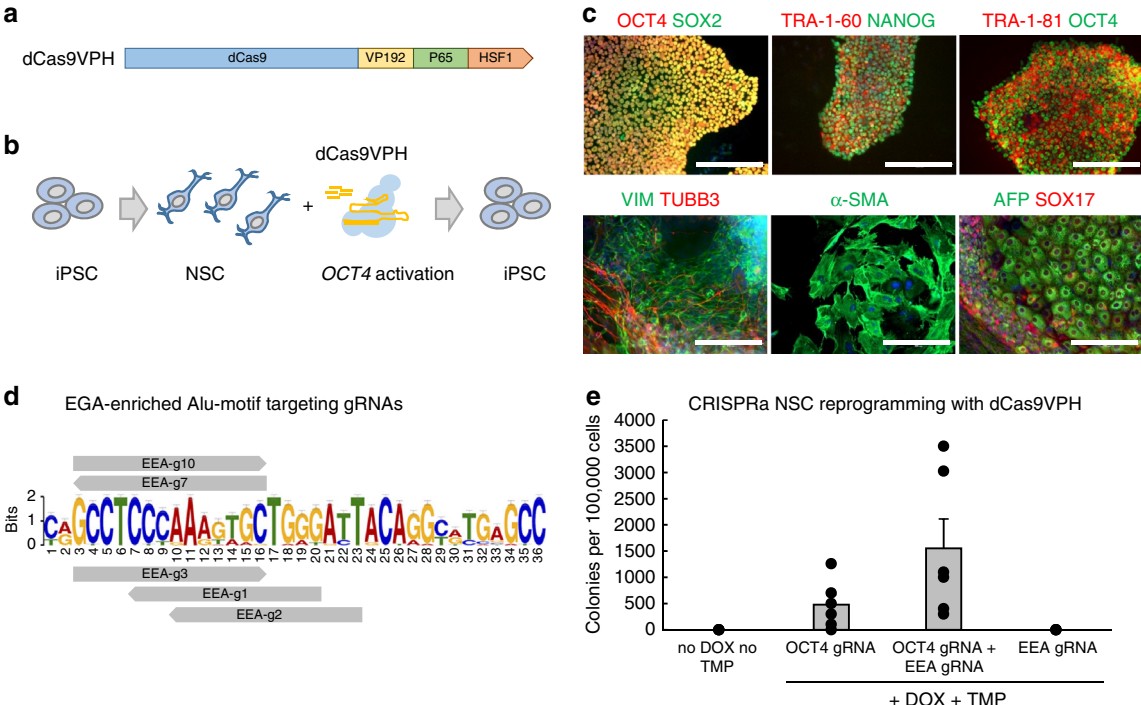

**Fig. 1** CRISPRa-mediated reprogramming of NSCs and EEA-motif targeting. **a** Schematic representation of dCas9VPH structure. **b** Schematic representation of NSC reprogramming into iPSCs with dCas9VPH mediated *OCT4* activation. **c** Immunocytochemical detection of pluripotency markers in NCS-derived iPSCs (top row) and tri-lineage differentiation in plated embryoid bodies (bottom row). Nuclei stained blue. Scale bar = 200 μm. **d** Targeting of EGA enriched Alu-motif with SpdCas9 gRNAs. **e** Quantification of iPSC-like alkaline phosphatase positive colonies induced from NSCs. $n = 6$ independent inductions ($P = 0.053$, *OCT4* targeting with EEA-gRNAs vs. without EEA-gRNAs). Data presented as mean ± s.e.m., two-tailed Student's *t*-test

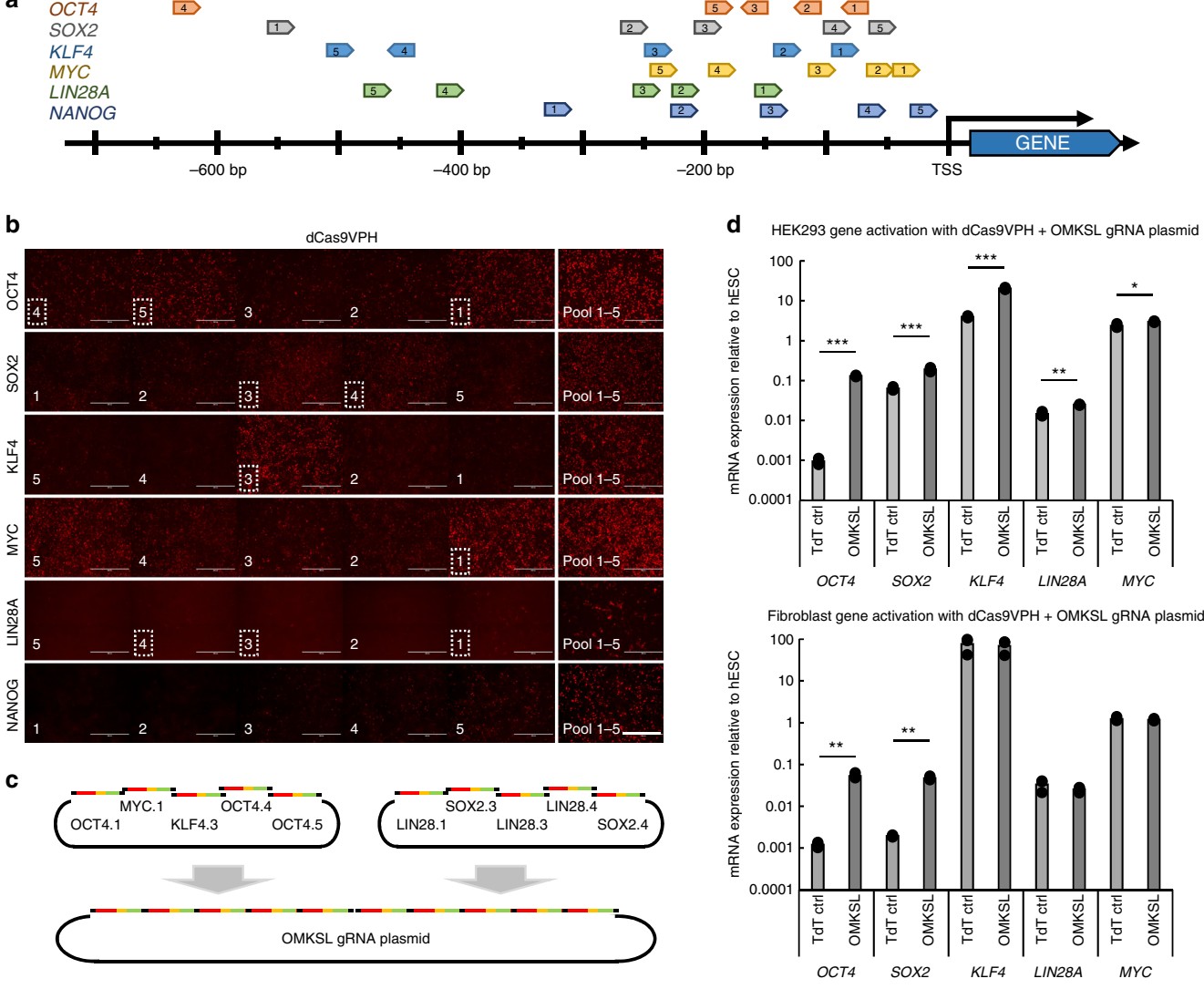

**Fig. 2** Optimization of dCas9 activator and gRNA targeting in HEK293 for reprogramming factor activation. **a** Locations of promoter targeting gRNAs for reprogramming factors (*OCT4, SOX2, KLF4, C-MYC, LIN28A*, and *NANOG*) in relation to transcription start site. **b** Immunocytochemical staining of reprogramming factors after single gRNA activation and pooled mixture of five guides in HEK293 with dCas9VPH. Pictures are in similar order to guides in Fig. 2a. Best performing guides used for plasmid cloning are marked with dotted lines. Scale bar = 400 μm. **c** Schematic representation of concatenated reprogramming factor gRNA plasmid construction. **d** Reprogramming factor activation by qRT-PCR, in HEK293 cells 3 days after transfection and HFFs 4 days after electroporation, using transiently expressed dCas9VPH effector. $n = 3$, data are from three independently treated samples. Data presented as mean ± s.e.m., two tailed Student's $t$-test. *$P < 0.05$, **$P < 0.01$, ***$P < 0.001$

targeting the 36 bp EEA consensus sequence (Fig. 1d). Addition of the EEA-motif gRNAs in the reprogramming mixture demonstrated a trend in increasing the number of alkaline phosphatase (AP) positive colonies ($P = 0.053$, Student's $t$-test) (Fig. 1e). This suggested that EEA-motif targeting could be useful for improving CRISPRa reprogramming efficiency.

**Pluripotency factor activation with CRISPRa**. To devise a reprogramming system for fibroblasts based solely on CRISPRa, we optimized the promoter targeting of single gRNAs to the canonical reprogramming factors *OCT4, MYC, KLF4, SOX2, LIN28A*, and *NANOG* in HEK293[24,25] (Fig. 2a, b). Best performing gRNAs targeting *OCT4, MYC, KLF4, SOX2*, and *LIN28A* (OMKSL) promoters were concatenated into a single plasmid and tested in transfected HEK293 and human foreskin fibroblasts (HFFs) with dCas9VPH activator (Fig. 2c, d). Robust activation of all targeted genes could be detected in HEK293, whereas HFFs demonstrated robust activation of *OCT4* and *SOX2* but not the

rest of the factors. This suggested that additional guides for *KLF4*, *LIN28A*, and *MYC* targeting would be required for efficient activation of these genes.

**Fibroblast reprogramming with CRISPRa**. Electroporation of primary skin fibroblasts with episomally replicating dCas9VPH plasmid, containing *TP53* targeting shRNA, EEA-motif targeting gRNA plasmid, reprogramming factor targeting gRNA plasmid (OMKSL), and an additional *KLF4* and *MYC* targeting gRNA plasmid (KM) resulted in the emergence of iPSC-like colonies (Fig. 3a). The resulting colonies could be expanded into iPSC lines demonstrating typical pluripotency markers and differentiation into three germ layer derivatives in vitro and in vivo (Fig. 3b and Supplementary Fig. 2a, b). These CRISPRa-induced iPSC lines presented normal karyotypes (Fig. 3c and Supplementary Fig. 2c), absence of transgenic vectors (Supplementary Fig. 2d), and clustered separately from HFFs, together with control Sendai virus-derived iPSCs (HEL46.11) and H9

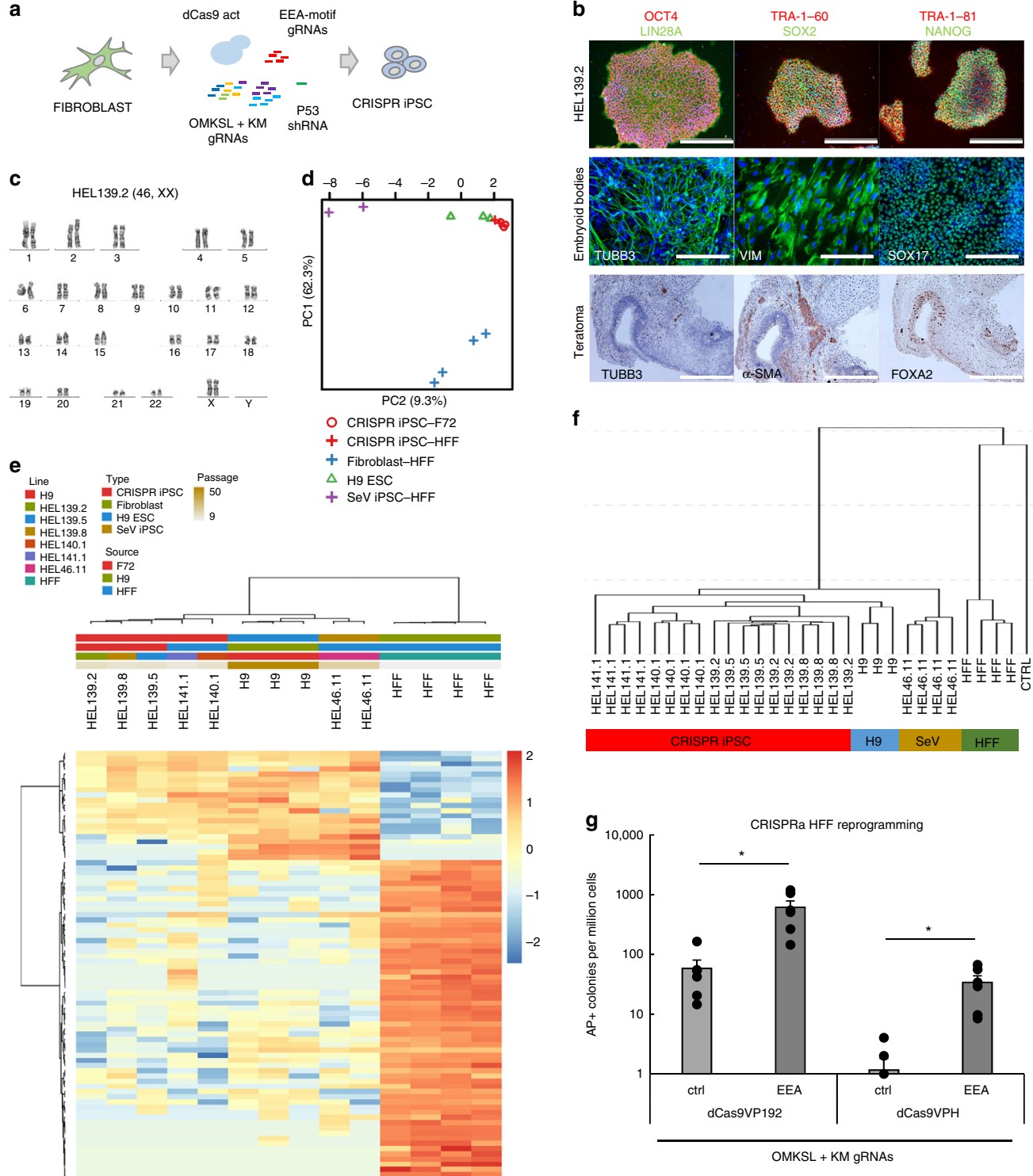

**Fig. 3** EEA-motif targeting enhances derivation of CRISPRa iPSCs from primary skin fibroblasts. **a** Schematic representation of skin fibroblast reprogramming with dCas9 activators. **b** Pluripotency factor expression in CRISPR-iPSC colonies (top row, scale bar = 400 μm) and tri-lineage differentiation markers for ectoderm (TUBB3), mesoderm (Vimentin and α-SMA), and endoderm (SOX17 and FOXA2) in embryoid bodies (middle row, scale bar = 200 μm) and teratomas (bottom row, scale bar = 800 μm). **c** Normal 46, XX karyotype of a CRISPRa iPSC line HEL139.2. **d** Principal component analysis of CRISPR iPSC lines, control PSC lines and HFFs based on expression of 123 significantly fluctuated genes. **e** Clustering of iPSC lines and HFFs based on expression of 85 significantly fluctuated and differentially regulated genes. **f** Clustering of CRISPR iPSC lines and control pluripotent stem cells based on DNA methylation. **g** Effect of VP192 and VPH domains and EEA-motif targeting on CRISPRa reprogramming efficiency of HFFs. $n = 6$ from three independent experiments. Data presented as mean ± s.e.m., two tailed Student's $t$-test. **$P < 0.01$

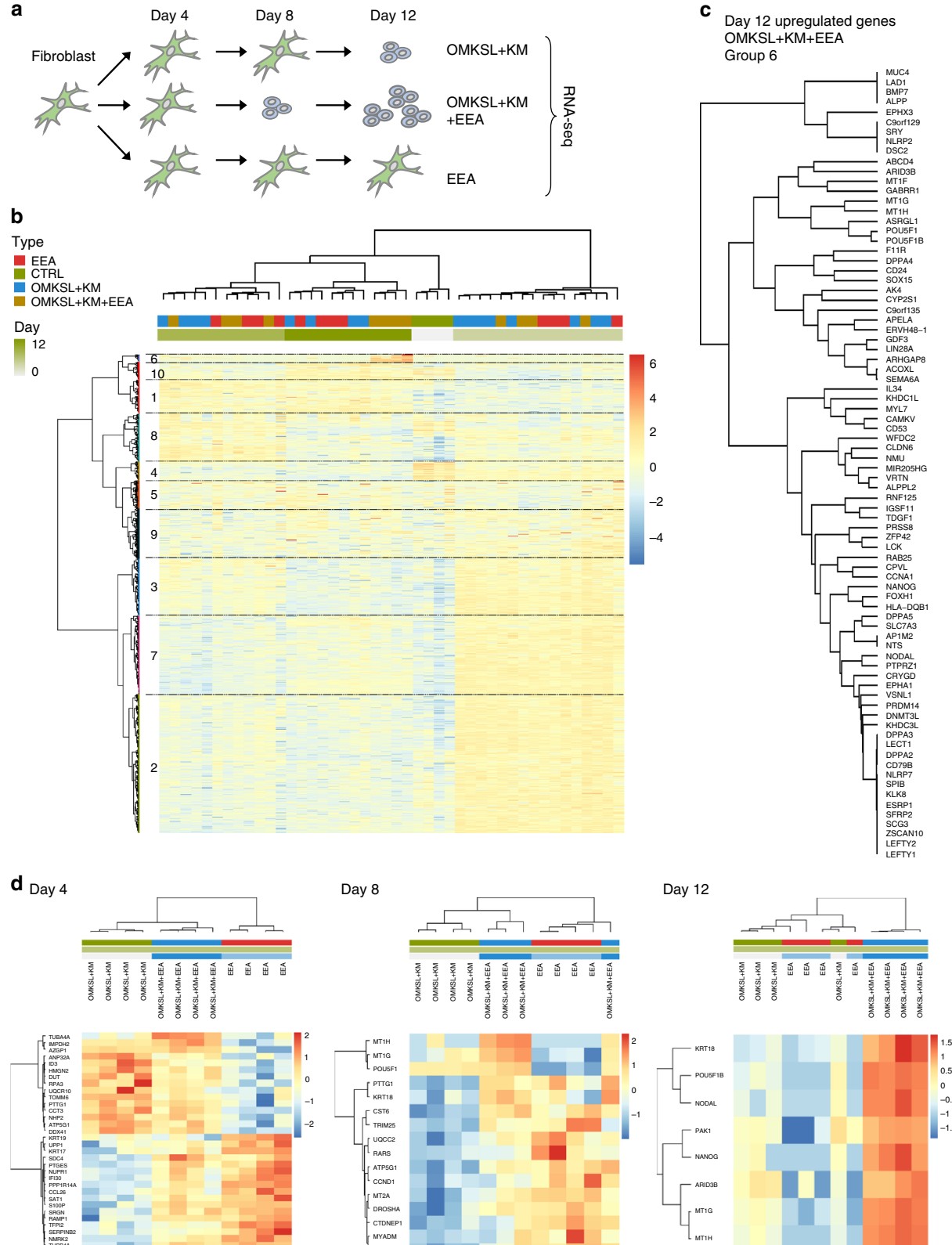

**Fig. 4** Transcriptional analysis of reprogramming cell populations. **a** Schematic representation of skin fibroblast reprograming with dCas9VP192 for RNA sequencing samples. **b** Clustering of all reprogramming samples based on expression of 4972 significantly fluctuated genes. Group 6 represents genes upregulated specifically by EEA-guides at day 12. **c** Upregulated genes at day 12 of reprogramming in OMKSL + KM + EEA targeted cells (Group 6). **d** Clustering of samples within day 4, 8, and 12 time points based on differentially regulated genes

embryonic stem cells, by transcriptional (Fig. 3d, e) and DNA methylation profiles (Fig. 3f). Overall, this demonstrated that CRISPRa reprogramming can be used to derive fully reprogrammed iPSCs from human skin fibroblasts.

In order to optimize the CRISPRa reprogramming method, we tested the effect of the activator domains and the EEA-motif gRNAs on reprogramming efficiency using HFFs. CRISPRa reprogramming using dCas9VP192 and EEA-motif gRNAs resulted in the most efficient AP positive colony formation (up to 0.062% of electroporated cells) (Fig. 3g). EEA-motif targeting greatly enhanced the CRISPRa reprogramming efficiency, ranging from 10.5-fold increase with VP192 ($P = 0.02$, Student's $t$-test) to 29.2-fold increase with VPH ($P = 0.02$, Student's $t$-test) (Fig. 3g). This effect was more prominent in CRISPRa reprogramming than in transgenic reprogramming of HFFs (Supplementary Fig. 3a). The dCas9VP192 activator performed better than dCas9VPH when targeting only *OCT4* for activation in an otherwise transgenic reprogramming approach (Supplementary Fig. 3b) or even when no gRNAs were present (Supplementary Fig. 3c). This suggested that the activator itself may interfere with the reprogramming process. We additionally tested P300 core fusions of the two activators, but they did not improve the reprogramming outcome (Supplementary Note 1 and Supplementary Fig. 4). As dCas9VP192 appeared to perform best in the CRISPRa reprogramming of human fibroblasts, it was used in the subsequent experiments.

**Transcriptional analysis of CRISPRa reprogramming.** To decipher the mechanism behind the increase in reprogramming efficiency mediated by the EEA-motif targeting, we conducted expression profile analysis of HFF cell populations undergoing CRISPRa-induced reprogramming in the presence and absence of the EEA-motif gRNAs (Fig. 4a). Based on fluctuated genes in the full data set, the samples clustered primarily by induction date (Fig. 4b). Additionally, by day 12 the OMKSL + KM + EEA gRNA treated cells clustered separately from the rest of the samples and demonstrated higher expression of 78 genes primarily associated with pluripotency and TGF-β signalling (Group 6 in Fig. 4b, c).

The bulk RNA samples represent heterogeneous cell populations where the majority of the cells are not undergoing complete reprogramming. The clustering seen by induction day may thus reflect nonspecific responses of the fibroblasts to handling, e.g., electroporation. Therefore, we also compared the samples within each time point. This revealed a common set of higher expressed genes on day 4 in the conditions containing EEA-motif targeting gRNAs (OMKSL + KM + EEA and EEA, Fig. 4d). These EEA-associated genes had a significantly higher number of EEA-gRNA 1 (EEA-g1) binding sites near their upstream regions ($-10$ kb to $+1$ kb from TSS) (mean 0.409 per kb, $n = 18$, $P = 5 \times 10^{-5}$, vs. genomic mean 0.215 per kb for protein coding genes, Monte Carlo sampling). This suggested a preferential initial activation of genes with multiple EEA gRNA target sites. A set of EEA-associated genes was also seen expressed higher in the day 8 samples (Fig. 4d), but these genes did not show enrichment for EEA-g1 sites (mean 0.280 per kb, $n = 13$, $P = 0.068$, Monte Carlo sampling). Significant enrichment was also not detected in the genes that were expressed higher by OMKSL + KM only on day 4 (mean 0.224 per kb, $n = 15$, $P = 0.314$, Monte Carlo sampling) (Fig. 4d), or in the pluripotency-associated genes that were expressed higher on day 12 (Group 6, mean 0.248, $n = 78$, $P = 0.070$, Monte Carlo sampling) (Fig. 4c). However, the day 8 higher expressed EEA-associated genes included factors like *TRIM25*, linked to LIN28A function[26], and *DROSHA* and *CCND1* which have been associated with cellular

reprogramming[27,28] (Fig. 4d). EEA-motif targeting at the mid stages of induction may thus contribute to more efficient expression of factors that can promote reprogramming, even if enrichment of EEA-g1 sites was not detected. Unlike day 4 and day 8 higher expressed genes, day 12 genes did not show division between EEA related and OMKSL + KM related sets (Fig. 4d), suggesting that the EEA-motif targeting primarily affects the initial stages of the reprogramming process prior to colony formation.

**NANOG and REX1 are EEA-associated reprogramming factors.** Detection of transcriptional changes occurring in small subsets of reprogramming cells can be challenging using RNA-seq of bulk mRNA. This was evident from the absence of detectable *LIN28A* reads from some of the sequencing samples, although LIN28A protein could clearly be detected by immunostaining in the forming colonies (Fig. 5a). This could also lead to poor detection of reprogramming factors targeted by EEA gRNAs. Assuming the EEA-associated reprogramming factors stay expressed in pluripotent cells, they should be detected more reliably in the day 12 samples, as the fibroblast background is diminished due to expansion of the reprogramming colonies (Fig. 5a). We therefore chose to test a set of seven pluripotency associated factors upregulated by day 12 in the OMKSL + KM + EEA reprogramming data set (Group 6) for their ability to enhance the reprogramming efficiency (Figs 4c and 5b). Transgenic expression of NANOG and REX1 in CRISPRa reprogramming in the absence of EEA-motif gRNAs, using optimized reprogramming factor gRNA plasmid (Supplementary Fig. 5), resulted in improved reprogramming efficiency (Fig. 5b). This indicated that NANOG and REX1 could be mediating the EEA-motif targeting effect. Assuming these factors are downstream effectors of EEA-motif targeting, their direct activation should also be enhanced by EEA-motif gRNAs. Accordingly, dCas9VP192 mediated activation of both *NANOG* and *REX1* promoters in transiently transfected HEK293 resulted in higher expression of these genes in the presence of the EEA-motif gRNAs compared with a TdTomato targeting control gRNA ($P = 0.009$ for *NANOG* and $P = 0.004$ for *REX1*, Student's $t$-test) (Fig. 5c). Both *NANOG* and *REX1* loci contain EEA-g1 binding sites near the genes (Supplementary Fig. 6a, b). *REX1* expression was improved by EEA-gRNAs even when *REX1* gRNAs were replaced with *NANOG* gRNAs, whereas *NANOG* expression was not improved by EEA-gRNAs in the presence of *REX1* gRNAs (Supplementary Fig. 6c). Therefore, *REX1* activation by targeting of the EEA-gRNA site near its promoter may represent a direct activation effect, possibly aided by NANOG mediated targeting of *REX1*. On the other hand, *NANOG* activation may be more dependent on additional reprogramming factors or *NANOG* promoter targeting by dCas9 activators. Both *NANOG* and *REX1* thus appear to be downstream targets of the EEA-motif gRNAs, contributing to its effect on improving reprogramming efficiency. Consistent with this, NANOG protein expression could be detected 2 days earlier in reprogramming colonies in the presence of the EEA-motif targeting gRNAs (Fig. 5d).

We additionally tested the rest of the reprogramming factors (*OCT4*, *SOX2*, *KLF4*, *MYC*, and *LIN28A*), and a set of non-pluripotency associated genes to determine if their activation is affected by simultaneous EEA-motif targeting in HEK293. However, this did not result in improved transcriptional activation (Supplementary Fig. 6d, e). The EEA-motif targeting thus appears specific only to a subset of reprogramming factors.

**Mechanism of EEA-motif targeting.** To further dissect the mechanism behind the EEA-motif targeting in CRISPRa

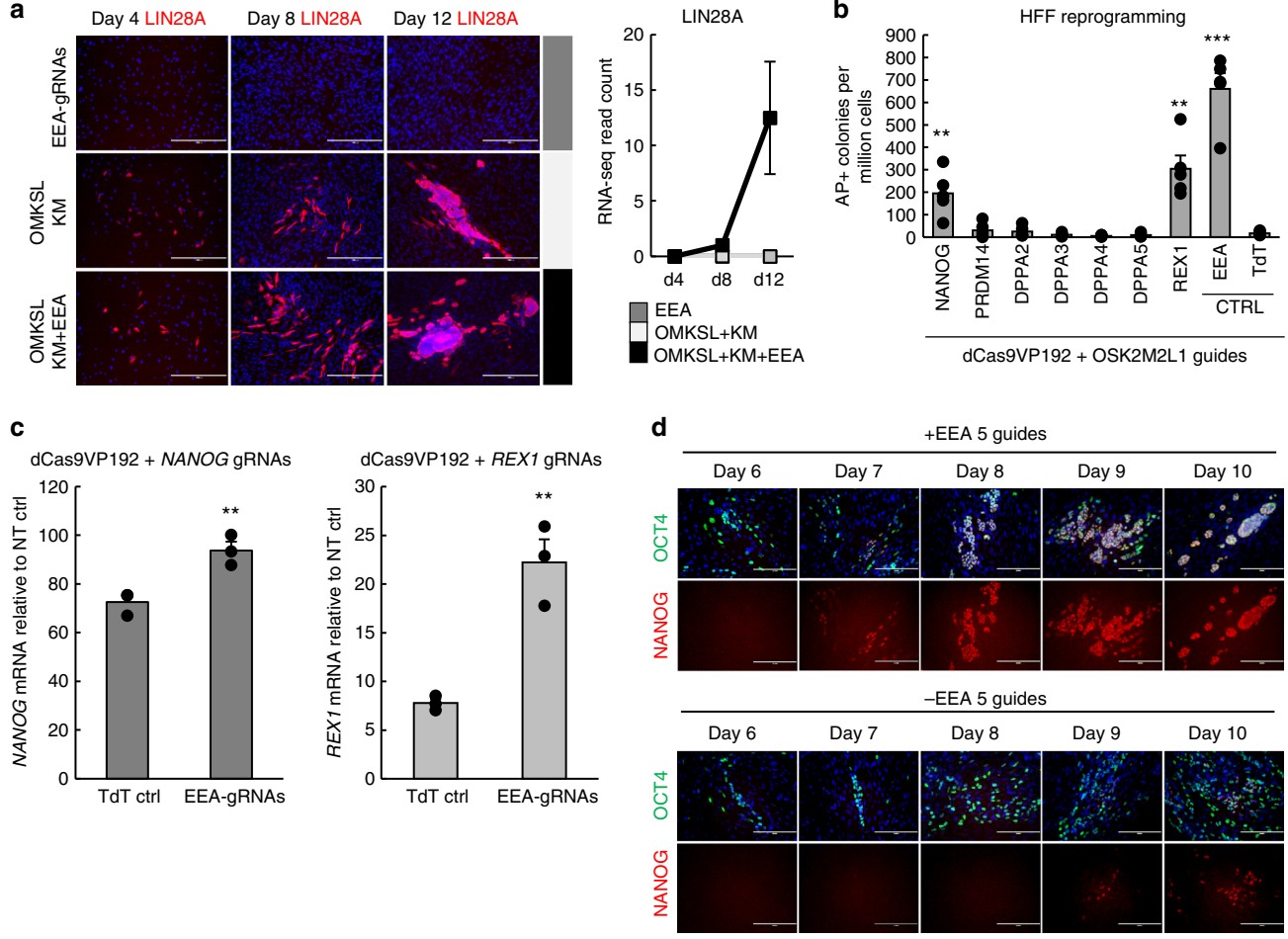

**Fig. 5** EEA-motif targeting improves *NANOG* and *REX1* activation. **a** Immunostaining of emerging iPSC colonies detects activation of targeted LIN28A before its mRNA reads become detectable in bulk RNA-seq. Scale bar = 400 μm. **b** HFF reprogramming efficiency after transgenic expression of additional factors belonging to Group 6 (Fig. 4c). $n = 5$, three independent experiments. **c** qRT-PCR quantification of *NANOG* and *REX1* activation using dCas9VP192 with and without EEA-motif gRNAs in HEK293. $n = 3$, three independent experiments. **d** NANOG activation can be detected by immunostaining in CRISPRa reprogramming colonies 2 days earlier in the presence of EEA-motif targeting guides. Scale bar = 200 μm. Data presented as mean ± s.e.m., two tailed Student's *t*-test. **$P < 0.01$, ***$P < 0.001$

reprogramming, we individually tested all the five guides targeting the EEA-motif. We also included control guides targeting common guide sequences found in human pluripotent stem cell super enhancers[29], to rule out nonspecific global DNA targeting. Of note, the most common gRNAs in these areas also contained multiple Alu sequences and the control guides 8, 9, and 10 also targeted parts of the EEA-motif. Reprogramming efficiency was mainly dependent on the EEA-motif guide 1 (Fig. 6a), which promoted as efficient reprogramming as the five guides together (Supplementary Fig. 7a). There was a noticeable difference in reprogramming efficiencies between EEA-g1 and EEA-g2, which are located next to each other in the EEA-motif consensus (Fig. 1e). These differences may be explained by guide nucleotide composition, as EEA-g2 contained multiple PAM proximal nucleotides that have been shown to be disfavoured[30]. In accordance with this, EEA-g1 activated a reporter construct with the EEA-motif consensus sequence with higher efficiency than EEA-g2 ($P = 0.02$, Student's *t*-test) (Fig. 6b, c and Supplementary Fig. 7b)[31]. The gRNA nucleotide sequence affecting its efficiency may thus be a crucial determinant in the EEA-motif targeting effect.

As the EEA-motif is located in the left arm of the Alu consensus sequence, we also assessed the expression of Alu sequences that could be detected in the STRT RNA-seq data in

the pluripotent stem cells and the reprogramming samples. Alu expression was higher in pluripotent cells than in fibroblasts (Fig. 6d). In reprogramming samples Alu expression peaked initially in all day 4 samples, possibly as a response to electroporation, and thereafter decreased (Fig. 6d). Alu expression rose again in the day 12 samples of the OMKSL + KM + EEA samples with higher numbers of pluripotent cell colonies. Overall, EEA-motif targeting did not appear to affect Alu expression in the reprogramming cell batches. Therefore, Alu expression itself is unlikely to explain the potentiating effect of EEA-motif targeting on reprogramming.

We next tested the impact of different dCas9-fused effector domains on the EEA-g1 effect in conventional reprogramming with transgenic OCT4, SOX2, KLF4 LIN28, and L-MYC. Interestingly, the effector domain did not have a significant impact on the reprogramming efficiency, whereas absence of the dCas9 protein resulted in reduced reprogramming efficiency compared to the dCas9VP192 control ($P = 0.007$, Student's *t*-test) (Fig. 6e). The effect of the EEA-motif targeting therefore appears to be mediated specifically by dCas9. It is possible, that dCas9 binding to high affinity guide sites in the EEA-motif may disrupt the chromatin locally to mediate more efficient activation of adjacent genes. As dCas9 binding to DNA has been demonstrated to open chromatin near its target site[32], we performed ATAC-seq

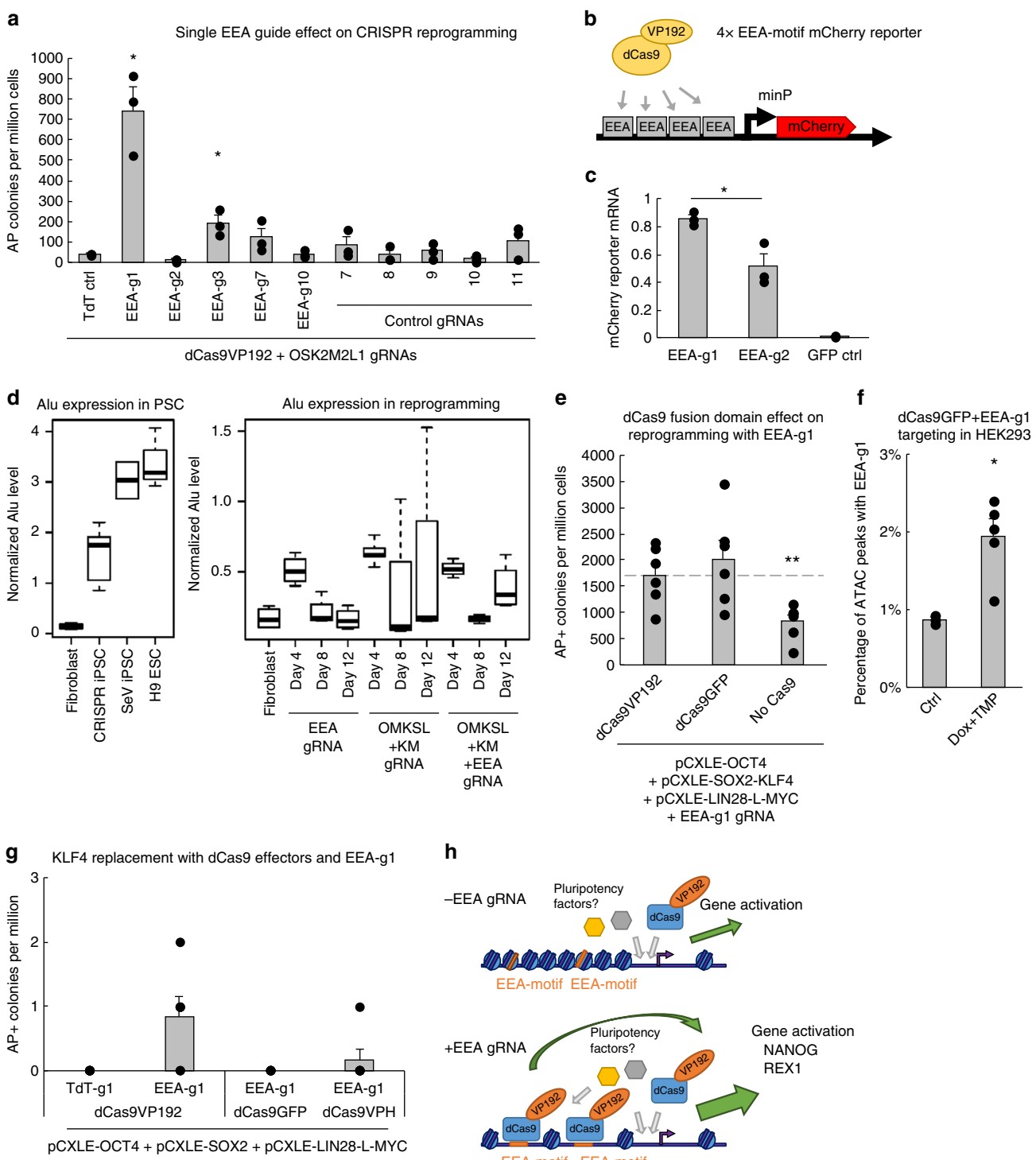

**Fig. 6** EGA-enriched Alu-motif targeting is associated with activation and chromatin opening. **a** Effect of single EEA-motif targeting gRNAs (Fig. 1e) on CRISPRa reprogramming efficiency of HFFs. $n = 3$, data are from three independent experiments. **b** Schematic representation of EEA-motif reporter plasmid activation. **c** qRT-PCR quantification of mCherry reporter activation with EEA-motif gRNAs 1 and 2 in reporter transfected HEK293 relative to five gRNAs. EEA-g1 activates the reporter with higher efficiency than EEA-g2 ($P = 0.02$). $n = 3$, three independent experiments. **d** Expression of Alu derived transcripts in STRT-seq data of pluripotent stem cells and CRISPRa reprogramming cell populations. Centre line, box, and whiskers represent median, quartiles, and 1.5 IQR, respectively. **e** Effect of dCas9 fusion domain on reprogramming efficiency in transgenic reprogramming with OCT4, SOX2, KLF4, LIN28, L-MYC, and EEA-g1. $n = 6$, data are from three independent experiments. **f** Increase in the presence of EEA-g1 sequence in ATAC-seq peaks in TetON-DDdCas9GFP and EEA-g1 expressing HEK293. ctrl $n = 3$, DOX + TMP $n = 5$. **g** Effect of dCas9 fusion domain on reprogramming efficiency in transgenic reprogramming with OCT4, SOX2, LIN28, L-MYC, and EEA-g1. $n = 6$, data are from three independent experiments. **h** Schematic model of EEA-motif targeting in gene activation and reprogramming. Data presented as mean ± s.e.m., two tailed Student's $t$-test. *$P < 0.05$, **$P < 0.01$

on samples of HEK293 cells expressing the EEA-g1 and a DOX- and TMP- inducible version of the DDdCas9GFP protein. DOX and TMP treated samples were found to have increased percentage of peaks with overlapping EEA-g1 sites (0.8% in controls, $n = 3$, vs. 1.9% in treated cells, $n = 5$, $P = 0.011$, Student's $t$-test) (Fig. 6f). This supports a mechanistic model in which dCas9 binding to high efficiency guide sites in the EEA-motif can lead to interference of the local chromatin structure near these elements, which may then contribute to the reprogramming process. As the EEA-gRNA-mediated increase in reprogramming efficiency is weaker in transgenic reprogramming than in CRISPRa reprogramming (Fig. 3g and Supplementary Fig. 3a), we additionally tested the EEA-g1 in suboptimal transgenic reprogramming conditions in the absence of transgenic KLF4. This resulted in complete abrogation of AP positive colony formation except in the presence of dCas9 activators targeting the EEA-motif (Fig. 6g and Supplementary Fig. 7c). Therefore, the activation function of the different dCas9 effectors appears to be important for the CRISPRa reprogramming process, since colonies only formed in dCas9VP192 and dCas9VPH containing conditions. This could be explained by improved activation of pluripotency factors with abundant EEA-motif sequences near them (Fig. 6h). These factors, such as NANOG, may help overcome the absence of KLF4 overexpression, as has been demonstrated by the use of alternative transgenic human reprogramming factor combinations (e.g., OCT4, SOX2, NANOG, and LIN28A)[25].

Finally, we tested if the fibroblast CRISPRa reprogramming system could be transferred into inducible transposon-based vectors. To this end, we inserted the DDdCas9VP192 activator, under a DOX-inducible promoter, into a PiggyBac vector and the OSK2M2L1 cassette and five guides for the EEA-motif into another one (Supplementary Fig. 5a and Supplementary Fig. 8a). HFFs electroporated with the PiggyBac vectors formed AP positive colonies, which could be expanded into stable iPSC lines, differentiated into three embryonic germ layer derivatives, and re-induced upon DOX and TMP addition (Supplementary Fig. 8). This demonstrated the applicability of the CRISPRa reprograming system in primary cells using various plasmid vectors and the option for the establishment of secondary reprogrammable systems based on CRISPRa.

## Discussion

CRISPR activator approaches hold great potential for controlling cellular reprogramming. The high multiplexing capacity of the system allows simultaneous targeting of large numbers of endogenous genes and genomic control elements using only short guide RNA molecules. This type of approach, combined with large scale synthesis of nucleic acids, can enable comprehensive targeting of gene regulatory networks with great precision for controlling cellular fate. However, until now no robust methods have been described for reprogramming human cells into pluripotency by CRISPRa.

We present a method for the efficient conversion of primary human fibroblasts into bona fide iPSCs based entirely on the transcriptional control of endogenous genes by CRISPRa. Activation of core reprogramming factor promoters alone was sufficient but inefficient, whereas additional targeting of a common Alu element brought the efficiency close to established reprogramming methods (Supplementary Fig. 5c). The more complex activator domains did not improve reprogramming efficiency, which mirrors previously reported results for gene activation[33], and suggests that the benefit of simple additional fused activation domains may be limited.

It has been estimated that 13% of human genome consists of Short Interspersed Nuclear Element (SINE) sequences, including Alu elements[34]. Accordingly, EEA-g1 sites can be found in more than 360,000 sites in the human genome. Due to the high abundance of EEA-g1 sites, the motif itself would not be expected to mediate a very strong or specific effect. This is also apparent from the inability of the EEA-gRNAs to reprogram cells by themselves. However, enrichment of the motif sequences near genes may end up enhancing the effect of the motif, as seen in the higher expression of the genes at day 4 that contain multiple EEA-motifs. Although we did not detect significant enrichment of EEA-g1 sites near the 78 pluripotency associated genes upregulated at day 12 (Group 6), Alu family repeats in general have been reported to be enriched near promoters of pluripotent stem cell expressed genes[35]. In the reprogramming context, this may end up biasing the EEA-motif targeting to preferentially affect pluripotency factors. However, in bulk RNA data this effect may be masked by the background of non-reprogramming cells, and therefore more thorough characterization of the EEA-motif targeting effect will require enrichment for the cell populations that undergo successful reprogramming[36,37].

The EEA-motif targeting effect on reprogramming appears to be associated with dCas9 effector-mediated opening of local chromatin, that promotes pluripotency factor activation. Alternatively, it is possible that dCas9 binding to the EEA-motif interferes with other possibly repressive factors targeting the motif. Alu elements have been linked with insulator function, including those near KRT18, which was upregulated at day 8 by EEA-motif targeting (Fig. 4d)[38]. Therefore, the opening of these elements, or interference with their function, may contribute to more efficient activation of nearby genes by interfering with chromatin insulation. Further insight into the mechanisms will require characterisation of factors binding to the motif during reprogramming as well as characterization of the function of early embryo factors, which are known to target the motif, e.g., PRD-like totipotent cell homeodomain factors and HNF4α[31,39–42].

In conclusion, CRISPRa reprogramming will provide a powerful tool for inducing pluripotent cells. The core method described here can be further improved by targeting known pluripotency genes and regulatory elements, as well as by screening for novel reprogramming factors and elements[23,43,44]. This will pave way for the development of more comprehensive CRISPRa reprogramming strategies, which in combination with transgenic factors, RNAi, and small molecular compounds, will promote more efficient and specific reprogramming of human cells for future applications.

## Methods

**Ethical consent**. The generation of the human induced pluripotent stem cell lines used in this study was approved by the Coordinating Ethics Committee of the Helsinki and Uusimaa Hospital District (Nro 423/13/03/00/08) with informed consent of the donors.

**Cell culture**. HEK293 cells (ATCC line CRL-1573), human foreskin fibroblasts (HFFs, ATCC line CRL-2429), and adult human dermal fibroblasts (derived in house) were cultured in fibroblast medium (Dulbecco's modified Eagle's medium (DMEM; Life Technologies) containing 10% fetal bovine serum (FBS; Life Technologies), 2 mM GlutaMAX (Life Technologies), and 100 μg/ml penicillin-streptomycin (Life Technologies)). Human induced pluripotent cells and embryonic stem cells were cultured on Matrigel (BD Biosciences) coated plates in E8 medium (Life Technologies) and split using 0.5 mM EDTA. Medium was changed every other day. All cells were kept in an incubator at 37 °C and 5% $CO_2$ and tested negative for mycoplasma.

**Guide RNA design and production**. Guide RNAs were designed and assembled as described by Balboa et al.[12]. Briefly, guide RNA expression cassettes, containing U6 promoter, chimeric single guide RNA and a Pol III terminator were assembled by PCR and concatenated into plasmids using Golden Gate assembly. Concatenated guide sets were cloned into episomal OriP-EBNA1 containing plasmids for

reprogramming experiments. A list of guide RNA oligonucleotides is provided in the Supplementary Table 1.

**dCas9 activator plasmid construction**. dCas9VPH construct was cloned by adding a P65-HSF1 containing fragment from lenti-MS2-P65-HSF1_Hygro (gift from Feng Zhang, Addgene Plasmid #61426) after the VP192 domain by PCR. dCas9VPP300 was cloned by PCR amplifying the P300 core domain from human cDNA and cloning it after the VP192 domain, as described by Hilton et al.[16]. dCas9VPPH was cloned by adding the P65-HSF1 domain in fusion after the VP192-P300 core domain. Activator plasmids were first cloned into CAG-dCas9VP192-T2A-GFP-IRES-Puro backbone and further cloned into pCXLE-dCas9VP192-T2A-EGFP-shP53 (Addgene plasmid #69535) backbone with XhoI and BsrGI. Plasmids used in this study will be made available on Addgene https://www.addgene.org/Timo_Otonkoski/ see also Supplementary Table 2.

**Cell transfection**. HEK293 cells were seeded on tissue culture treated 24 well plates 1 day prior to transfection ($10^5$ cells/well). Cells were transfected using 4:1 ratio of FuGENE HD transfection reagent (Promega) in fibroblast culture medium with 500 ng of dCas9 transactivator encoding plasmid and 100–200 ng of guide RNA-PCR or 250 ng of dCas9 transactivator encoding plasmid and 250 ng of concatenated guide RNA encoding plasmid. Cells were cultured for 72 h post-transfection, after which samples were collected for qRT-PCR or immunocytochemical staining. HEK293 cells containing the destabilized dCas9 activators and guides were transfected with dCas9 activator, guide RNA and PiggyBac transposase plasmids, 100 ng of each, and selected with Puromycin (2 µg/ml; Sigma) and G418 (0.5 mg/ml; Life Technologies).

**Quantitative reverse transcription PCR**. Total RNA was extracted from cells using NucleoSpin Plus RNA kit (Macherey-Nagel). RNA quality and concentration was measured by spectrophotometry using SimpliNano (General Electric). One microgram of total RNA was denatured at 65 °C for 1 min and used for reverse transcription (RT) with 0.5 µL Moloney murine leukemia virus (MMLV) reverse transcriptase (M1701, Promega), 0.2 µL Random Primers (C1181, Promega), 1 µL Oligo(dT)18 Primer (SO131, ThermoFIsher) and 0.5 µL Ribolock RNAse inhibitor (EO0382, ThermoFisher) for 90 min at 37 °C. For qRT-PCR reactions, 50 ng of retrotranscribed RNA were amplified with 5 µL of forward and reverse primer mix at 2 µM each using 5× HOT FIREPol EvaGreen qPCR Mix Plus (no ROX) in a final volume of 10 µL. QIAgility (Quiagen) liquid handing system was used for pipetting the reactions into 100 well disc that were subsequently sealed and run in Rotor-Gene Q (Qiagen) with a thermal cycle of 95 °C for 15 min, followed by 40 cycles of 95 °C, 25 s; 57 °C, 25 s; 72 °C, 25 s, followed by a melting step. Relative quantification of gene expression was analysed using ΔΔCt method, with cyclophilin G (*PPIG*) as endogenous control and an exogenous positive control used as calibrator. Expression levels are relative to non-treated cells or to hESC as indicated in the figure legends. A list of primers used is provided in the Supplementary Table 3.

**NSC differentiation**. Human NSCs were derived by differentiating human iPSC HEL24.3[45], and HEL46.11 lines using small molecule cocktail as described elsewhere[46], with minor adjustments. Briefly, iPSCs were detached with StemPro Accutase (Thermo Fisher Scientific) and dissociated gently into single cells suspension in hES-medium in the presence of 5 µM ROCK inhibitor (ROCKi; Y-27632, Selleckchem), 10 µM SB431542 (SB; S1067, Selleckchem), 1 µM dorsomorphin (DM; P5499-5MG, Sigma), 3 µM CHIR-99021 (CHIR; Tocris) and 0,5 µM purmorphamine (PMA; 04-0009, Stemgent) After 2 days, medium was changed to N2B27 medium (DMEM/F12:Neurobasal (1:1) supplemented with N2 and B27 without vitamin A, NEAA, PenStrep (all Thermo Fisher Scientific) and heparin (2 µg/ml; H3149-50KU, Sigma)) containing the same small molecule cocktail as above. On day 4, SB and DM were withdrawn and 150 µM ascorbic acid (AA) was added to N2B27. On day 6, the neurospheres were dissociated with 1 ml pipette and plated on Matrigel in N2B27 media containing AA, CHIR and PMA (growth media). First two passages were split at 1:3 ratio and cells were plated into growth media containing 5 µM ROCKi, which was removed next day. Later passages were split with 1:10 and 1:20 ratio using StemPro Accutase. Media were changed every other day.

**NSC reprogramming**. NSCs were grown for at least five passages before electroporation. For electroporation, cells were detached with StemPro Accutase and dissociated into single cells. Cells were washed once with PBS and electroporated with Neon Transfection system (Invitrogen). Two million cells were used per electroporation using 100 µl tips with 1300 V, 30 ms, one pulse settings. A quantity of 2 µg of PB-tight-DDdCas9VPH-GFP-IRES-Neo activator plasmid, 1.5 µg PB-GG-OCT4-1-5-PGK-Puro gRNA plasmid, and 1.5 µg PB-GG-EEA-5g-PGK-Puro gRNA plasmid were used with 0.5 µg PiggyBac rtTA and 0.5 µg of PiggyBac transposase plasmids. One million electroporated cells were plated per 35 mm plate coated with Matrigel in N2B27 media supplemented with 5 µM ROCKi and 10 ng/ml of basic FGF (bFGF, PeproTech). ROCKi was removed the next day. Two days after electroporation cells were treated with Puromycin (0.5 µg/ml; Sigma) and G418 (200 µg/ml; Life Technologies) for 5 days. On day 8 after electroporation

reprogramming was initiated by adding doxycycline (DOX, 2 µg/ml; Sigma) and trimethoprim (TMP, 1 µM; Sigma). After 5 days of induction media was changed to hES-medium gradually over a week. During the conversion process cells were split 3 times. On day 18 of induction cells were fixed with 4% paraformaldehyde (PFA) for AP staining or picked for iPSC derivation. Media were changed every other day.

**Fibroblast reprogramming**. Human skin fibroblasts were detached as single cells from the culture plates with TrypLE Select (Gibco) and washed with PBS. Cells were electroporated using the Neon transfection system (Invitrogen). A total of $10^6$ cells and 6 µg of plasmid mixture, containing 2 µg of dCas9 activator plasmid and 4 µg of guide plasmids, were electroporated in a 100 µl tip with 1650 V, 10 ms, and 3× pulse settings. Electroporated fibroblasts were plated on Matrigel coated 100 mm diameter cell culture plates in fibroblast medium. After 4 days cell culture medium was changed to 1:1 mixture of fibroblast medium and hES-medium (KnockOut DMEM (Gibco) supplemented with 20% KO serum replacement (Gibco), 1% GlutaMAX (Gibco), 0.1 mM beta-mercaptoethanol, 1% nonessential amino acids (Gibco), and 6 ng/ml basic fibroblast growth factor (FGF-2; Sigma)) supplemented with sodium butyrate (0.25 mM; Sigma). When first colonies started to emerge, cell culture medium was changed to hES-medium until colonies were picked. For iPSC line derivation, colonies were picked manually and plated on Matrigel coated wells in E8 medium. Media were changed every other day. Reprogramming with transgenic transcription factors was performed as described elsewhere[47] using pCXLE-OCT3/4-shP53, pCXLE-hSK, pCXLE-hUL (Addgene plasmid #27077, #27078, #27080) and pCXLE-OCT4 or pCXLE-SOX2 derived from pCXLE-OCT3/4-shP53 and pCXLE-hSK, respectively. For PiggyBac reprogramming, HFFs were electroporated as described above with PB-tight-DDdCas9VP192-GFP-IRES-Neo, PB-CAG-rtTA-IRES-Neo, PB-GG-EEA-5g-OSK2M2L1-PGK-Puro, and PiggyBac transposase plasmids. Electroporated cells were plated on cell culture dishes in fibroblast medium. Five days after electroporation cells were selected with Puromycin (1 µg/ml; Sigma) and G418 (0.5 mg/ml; Roche) for 2 days after which the selection antibiotic amounts were halved. Selected cells were induced as described above in the presence of TMP (1 µM) and DOX (2 µg/ml). Fresh DOX was supplemented daily. For RNA sequencing samples, passage 10 foreskin fibroblasts were electroporated with pCXLE-dCas9VP192-GFP-shP53 and combinations of GG-EBNA-OSKML-PP, GG-EBNA-KM-PP, and GG-EBNA-EEA-5guides-PP plasmids. A total of 250k (day 4 samples) to 125k cells (day 8 and 12 samples) were plated on Matrigel coated six-well culture plates per well and induced as described above.

**Pluripotent cell line derivation**. HEL139 clones were derived from adult female skin fibroblasts (F72) using GG-EBNA-OMKSL-PP, EBNA-EEA-5guides-PGK-Puro, and an additional GG-EBNA-KM-PP plasmid (KLF4 and MYC five guides each). HEL140 was derived from neonatal male skin fibroblasts (HFFs) with GG-EBNA-OMKSL-PP, EBNA-EEA-5guides-PGK-Puro, and GG-EBNA-KM-PP plasmids. HEL141 was derived from neonatal male skin fibroblasts (HFFs) with GG-EBNA-EEA-5guides-PGK-Puro, GG-EBNA-KM-PP, and GG-EBNA-OS-PP plasmids (OCT4 and SOX2 five guides each). The above cell lines were derived using pCXLE-dCas9VPH-T2A-GFP-shP53 activator plasmid. HEL144 was derived from neonatal male skin fibroblasts (HFFs) with inducible PiggyBac vectors using PB-tight-DDdCas9VP192-GFP-IRES-Neo activator and PB-EEA-5g-OSK2M2L1-PGK-Puro guides. Control cell line HEL46.11 was derived from HFFs using CytoTune Sendai Reprogramming Kit (Life Technologies) according to manufacturer's instructions.

**Alkaline phosphatase staining**. iPSC colonies were fixed with 4% paraformaldehyde (PFA) solution for 10 min and washed with phosphate buffered saline (PBS). Thereafter cells were stained in NBT/BCIP (Roche) containing buffer (0.1 M Tris HCl pH 9.5, 0.1 M NaCl, 0.05 M $MgCl_2$) until precipitate developed. Reaction was stopped by washing the plates with PBS.

**Immunocytochemistry**. Cells were fixed with 4% PFA, permeabilized using 0.2% Triton X-100, and treated with Ultra Vision block (ThermoFisher). Primary antibodies were diluted in 0.1% Tween-20 PBS and incubated either overnight at 6 °C with the given dilutions or 2 days in 6 °C with halved primary antibody amounts. Secondary antibody incubations were done in room temperature for 30 min in the presence of Hoechst33342 to stain the nuclei. Primary antibodies used were: LIN28A (1:250, D84C11 and D1A1A, Cell Signaling), NANOG (1:250, D73G4, Cell Signaling), OCT4 (1:500, sc-8628, Santa Cruz), SOX2 (1:250, D6D9, Cell Signaling), KLF4 (1:250, HPA002926, Sigma-Aldrich), C-MYC (1:250, D3N8F, Cell Signaling; 1:250, [Y69] ab32072, Abcam), TRA-1-60 (1:50, MA1-023, ThermoFisher), TRA-1-81 (1:100, MA1-024, ThermoFisher) TUBB3 (1:500, MAB1195, R&D Systems), AFP (1:400, A0008, Dako), SMA (1:200, A2547, Sigma), VIMENTIN (1:500, sc-5565, Santa Cruz). SOX17 (1:500, AF1924, R&D Systems), acetyl Histone 3 (1:500, ab47915, Abcam). Secondary antibodies used were: AlexaFluor 488: donkey anti-goat (1:500, A11055 and 11058; Invitrogen), donkey anti-mouse (1:500, A21202 and A21203; Invitrogen) and donkey anti-rabbit (1:500, A21206 and A21207; Invitrogen).

**Embroid body assay.** iPSCs were split into small clumps and plated on low attachment dishes (Corning) in hESC medium without bFGF to allow embroid body (EB) formation. The EB culture medium was supplemented overnight with 5 µM ROCK inhibitor (Y-27632, Selleckchem) after the initial cell plating to improve cell viability. Medium was changed every other day. EBs were grown in suspension for 14 days, after which they were plated on gelatin coated cell culture dishes. EBs were allowed to form outgrowths for 7 days after which cells were fixed with 4% PFA for 30 min and permeabilized using 0.2% Triton X100 (Sigma) in PBS for 30 min. Fixed and permeabilized EB outgrowths were stained as described above.

**Teratoma assay.** About 200,000 morphologically intact iPSCs at passage 23 were intratesticularly injected into male NMRI nude mice (Scanbur). The resulting tumours were collected 2 months after injection, fixed with 4% PFA, and hematoxylin and eosin stained. Animal care and experiments were approved by the National Animal Experiment Board in Finland (ESAVI/9978/04.10.07/2014).

**DNA methylation assay and analysis.** For DNA methylation array samples, passage 11-20 CRISPRa iPSCs, passage 35 SeV iPSCs, and passage 50 H9 ESCs were collected, one well of six-well plate each, as well as HFF control fibroblasts. DNA was purified using DNeasy Blood & Tissue Kit (Qiagen) and the concentrations were adjusted to 11 ng/µl using Qubit assay (Thermo Fisther Scientific). PicoGreen Assay (ThermoFisher) was used for subsequent normalization of samples. DNA of the samples and three controls (Zymo_low, Zymo_high and 1331-1 CEPH) was treated with sodium bisulphite using the EZ DNA methylation kit (Zymo Research). DNA methylation was quantified using the Illumina Infinium HumanMethylationEPIC BeadChip on an Illumina iScan System using the manufacturer's standard protocol. Raw IDAT files were processed with Illumina's GenomeStudio v2011.1, and normalized beta-values, except two controls (Zymo_low and Zymo_high), were applied for the clustering of the DNA methylation profile.

**STRT-sequencing.** RNA samples were collected in Trizol Reagent (Life Technologies), 100 µl of chloroform per 500 µl of sample was added and mixed with Trizol Reagent. After centrifugation (12,000 g for 15 min) the transparent upper phase was collected and RNA was further purified using NucleoSpin RNA kit (Macherey-Nagel). RIN values were measured by Bioanalyzer (Agilent) and total-RNA concentrations scaled to equal (10 ng/µl) using Qubit assay (Thermo Fisher Scientific). Bulk-RNA transcriptome analysis was performed by the STRT RNA-seq method[48] with minor modifications[49]. Briefly, 10 ng of high-quality input RNA was converted to cDNA and amplified to form an Illumina-compatible 46-plex library. In total, 25 PCR cycles were used, but as six base-pair unique molecular identifiers (UMIs) were applied, only the absolute number of unique reads was calculated per analyzed sample. The library was sequenced on three lanes of Illumina HiSeq2000 instrument.

**STRT data analysis.** The sequenced raw STRT reads were processed by STRTprep[49]; v3dev branch, d7efcde commit (https://github.com/shka/STRTprep/tree/v3dev). In brief, redundant reads after demultiplexing were excluded according to UMI, and the nonredundant reads were aligned to hg19 human reference genome sequences, ERCC spike-in sequences, and human ribosomal DNA unit (GenBank: U13369). Uniquely mapped reads within (1) the 5′-UTR or the proximal upstream (up to 500 bp) of the RefSeq protein coding genes, (2) the 5′-UTR or the proximal upstream of some PRD genes, which were not yet defined by RefSeq[19], and (3) within the first 50 bp of spike-in sequences, were counted. The processed reads were aligned also to Alu canonical sequence (http://www.repeatmasker.org/AluSubfamilies/humanAluSubfamilies.html) using the same methodology with STRTprep, to count Alu transcripts.

Significance of fluctuation on gene expression was tested by comparison with fluctuation of spike-in levels as shown in ref.[49]. Principal component analysis (PCA) was performed on fluctuated genes (corrected p-value <0.05) via PCA plugin of STRTprep. Differential expression between the sample types were tested by SAMstrt[50]. Regulated genes were selected by corrected fluctuation p-value <0.05 (as significant degree of expression change) and differential expression q-value <0.05 (as significant contrast between the types). Hierarchical clustering of normalized expression profiles and the illustration were performed by aheatmap function in NMF package[51] via heatmap_diffexp plugin of STRTprep; color gradient in the heatmap represents Z-score of each gene, and the profile was clustered by Ward's algorithm on Spearman correlation based distance. The fluctuated genes can be independent from the sample types (e.g., batch, circadian, cell-cycle etc.), while the regulated genes are selected by differential expression between the sample types. Therefore, the unsupervised clustering of samples by fluctuated genes (PCA and the Fig. 4b) reflects the major differences among the samples, while clustering using the regulated genes aims for grouping based on these genes.

HEL136 cell line was excluded from sequencing analysis due to missing karyotype. One HEL46.11 control sample was excluded from sequencing analysis due to higher expression of differentiation associated genes, indicative of differentiated cells in the sample well.

**ATAC-sequencing.** HEK293 cell lines used for ATAC-seq samples were transfected with PiggyBac vectors for TetON-DDdCas9GFP-IRES-Neo, CAG-rtTA-IRES-Neo, and 36bp-guide1-PGK-Puro. Cells were selected with Puromycin and G418, after which cells were sub cloned from single cells and clones were selected based on homogenous GFP expression upon doxycycline addition. For sample preparation, cell clones were treated for 3 days with doxycycline (1 µg/ml) and trimethoprim (1 µM). Fifty thousand cells were collected for ATAC-seq library preparation according to Buenrostro et al.[52]. Briefly, nuclei were extracted by continuous centrifugations and chromatin was exposed for enzymatic tagmentation and 12-plex Illumina-compatible library. All fragments were amplified by Phusion hot-start polymerase using 12 cycles of PCR in total. The ATAC-libary was sequenced on one lane of Illumina HiSeq2000 instrument. Peak calling was done with Homer looking for histone like peaks, using sample 5 (non-treated control) as background.

**Statistical analysis.** All the reprogramming experiments were replicated in three independent experiments with duplicate samples if enough cells were available. NSC inductions were replicated in six independent experiments with single samples per experiment. All sample sizes are indicated in the corresponding figure legends.

Statistical analysis was performed as described in the figure legends. P-values of less than 0.05 were considered significant (*$P < 0.05$, **$P < 0.01$, ***$P < 0.001$). Estimating EEA-g1 enrichment near upstream regions of EEA associated genes was done by Monte Carlo sampling with $10^5$ random permutations of $n$ number of genes (as defined in the text) from a pool of 19,806 protein coding genes.

**Data availability.** RNA Sequencing, DNA methylation, and ATAC-seq data related to Figs. 3d–f, 4, and 6d–f are available on Array Express: "E-MTAB-6185," "E-MTAB-6186," "E-MTAB-6194," "E-MTAB-6195".

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

## Acknowledgements

We thank J. Saarimäki-Vire, S. Eurola, H. Grym, A. Laitinen, M. Salmela, Y. Novik, I. Fransson, and A. Damdimopoulos for technical assistance with the work and S. Vuoristo for comments on the manuscript. This work has been supported by the 3i Regeneration project (number 40395/13; a TEKES Large Strategic Research Opening), Jane and Aatos Erkko Foundation, Academy of Finland (No. 297466 and 312437, Center of Excellence in Stem Cell Metabolism), Sigrid Jusélius Foundation, Novo Nordisk Foundation, Instrumentarium Science Foundation, the Doctoral Program in Biomedicine at University of Helsinki, Knut and Alice Wallenberg Foundation (KAW2015.0096), SNIC through Uppsala Multidisciplinary Center for Advanced Computational Science (UPPMAX) under project b2014069, Bioinformatics and Expression Analysis core facility (BEA) and The Mutation Analysis Core Facility (MAF) at the Karolinska University Hospital.

## Author contributions

J.W. and D.B. was responsible for conceptualization, J.W., D.B. and M.B. for methodology, S.K. for software, J.W. and S.K. was involved in formal analysis, J.W., D.B., M.B., K.K., E.M.J. in investigation, J.W. for writing – original draft, all authors were involved in writing – review and editing; R.T., J.K., and T.O. in supervision, and J.K. and T.O. in funding acquisition.

## Additional information

**Competing interests:** The authors declare no competing interests.

