## [Peer Review File · Nature Communications]

Reviewers' comments:

Reviewer #1 (Remarks to the Author):

General comments:

The programmable CRISPR/dCas9 system has been an attractive tool for activation of endogenous genes (CRISPRa). Several previous attempts (including ours) were tried to apply CRISPRa as an alternative approach for iPSC reprogramming but without success, although activation of endogenous pluripotency genes was achieved to a certain degree.

The study by Jere Weltner and colleagues demonstrated for the first time that CRISPRa can be used to reprogram human fibroblasts into iPSCs. Although the efficiency achieved with CRISPRa is lower than traditional transgenic reprogramming method, the study provides a proof-of-concept on the application of CRISPRa in iPSC reprogramming. Furthermore, the discovery that EEA targeting by CRISPRa can enhance reprogramming efficiency provides useful knowledge and tool for iPSC reprogramming. In general, the whole study is rather thoroughly conducted with sufficient statistical power. However, for some experiment, critical controls are lack which weaken the conclusion of the study. Some results are rather confusing at its current form and required more clear explanations. In addition, there are a few critical points that should be addressed.

Specific comments:

1. Successful reprogramming is typically achieved in only a small portion of cells. In line 64-65 (Fig. 1b), although VPP300 and VPPH fusions achieve a heterogeneous OCT4 activation, the OCT4 signal of positive cells are stronger in those positive cells. For this experiment, firstly, flow cytometry should be conducted to analyze the OCT4 activation in a quantitative manner. Secondly, what is the reason that heterogeneous is a "negative" criteria for choosing VPH rather than VPP300 and VPPH? As commented in Line 61, robust activation of OCT is critical of the success of reprogramming. VPP300 will give the best reprogramming outcome, unless VPP300 creates some unexpected negative effects. P300 CORE has been used in previous study for programmable histone acetylation. However, has the authors checked whether overexpressing P300 CORE will create unspecific histone acetylation. e.g. using anti-histone acetylated antibodies?

Control staining is required as supplementary: (1) Un-transfected; (2) Transfected with the fusion protein only (+ and - TMP); (3) Transfected with a scrambled gRNA ((+ and - TMP).

2. All CRISPRa effector proteins contains a GFP, does this create problem for the immunostaining using green channel?

3. CRISPRa reprogramming of NPC: Although the system used is DOX and TMP inducible, it is critical to show that: (1) The CRISPRa-derived iPSCs are GFP negative. (2) It's important to show that these are real CRISPRa-derived iPSCs. E.g. by PCR validating the presence of CRISPRa effector gene in iPSC genomic DNA.

(3) The iPSC-derived NPCs used for reprogramming are generated by Sendai reprogramming, although this method has very low risk of transgene integration, I cannot find the information about transgene integration testing in their previous paper (Stem Cell Research (2015) 15, 254–262).

(4) What about the efficacy of CRISPRa-mediated OCT4 activation as compared to expressing transgene OCT4 in NPCs? Reprogramming efficiency between overexpressing transgene OCT4 VS. CRISPRa mediated activation of OCT4? OCT4 Gene expression level by qPCR?

(5) As I can see from the M&M, there are two iPSC-derived NPC lines used for testing. Do both lines give equally good reprogramming?

(6) What about none iPSC-derived NPCs?

(7) It's important to have results from two extra controls to conclude that "Line 69, CRISPRa mediated activation of OCT4 alone was sufficient to ...": (a) Expression CRISPRa effector only; (b) CRISPRa with scrambled gRNAs.

4. EEA targeting. Again, Fig 1h. GFP transfection is not the appropriate control to conclude that EEA-motif targeting could be useful for improving reprogramming efficiency. None-targeting scrambled gRNAs should be used to compare with (similar to what Figure 6a).

Fig 1e. EEA gRNAs used are truncated gRNAs, please state that in the article.

Also, it would be better to demonstrate that CRISPRa binds to the EEA consensus sequences. E.g. by Chip-seq.

Fig. 1. g-h. The experiment has been tested in 5 independent inductions. Are these from the same NPC line?

Fig. 1h. There were 8.7-fold increase in reprogramming efficiency (transgenic reprogramming + EEA CRISPRa), VPH used. How is this experiment compared to Figure 6e? (VP192).

5. Fibroblasts reprogramming: (1) what is the basal expression level of these five genes in HEK293 cells? It would be difficult to activate genes that are already highly expressed. For example, MYC should be highly expressed.

(2) Line 83-84, based on Figure 2d, it's a little bit hard to agree that all four effectors showed comparable gene activation potential. VPP300 apparently outcompetes all the other three. The MYC gene is highly expressed in HEK293 cells, (TPM value equal to approximately 386 according to published RNA-seq data), the activation data from this gene in HEK293 cells cannot contribute to the conclusion.

(3) Comments regarding: Reprogramming of fibroblasts into iPSC by CRISPRa. Line 85-97, Figure 3 and Supplementary Figure 1. (3.1). Figure S3c should not appear and cited before Figure S1. Line 90. Re-order the supplementary figures. (3.2). Figure S1c, Karyotyping lack clone HEL136.2.

(3.3). the reprogramming is based on plasmid electroporation, was antibiotic selection used to enriched transfected cells? (3.4). Figure S1d, EBNA PCR showed that there is integration in some iPSC clones. Plasmid integration is randomly break and inserted. Could it be possible that the EBNA PCR fails to detect the integration? What about PCR of dCas9?

(3.5). Figure 3d-e, legends for PCA do not corresponding to the plot. The PCA analysis is based on "fluctuated genes, 268" and "165 significantly DEG, for the heat map plot". If these genes are preselected based on comparison between iPSCs and fibroblasts, there is no doubt that the samples will be clustered separately. What about PCA analysis based on all genes?

(3.6) Figure 3g. What is used as control (ctrl)? dCas9 activator with OMKSL gRNAs?

(4) Expressing dCas9VPH alone and transgenic factors decreased reprogramming efficiency. Figure S2 is normalized and compared to VP192. What is the reprogramming efficiency when comparing to transgenic factors only? Figure S2a. Controls required: Control 1: Transgenic reprogramming with transgenic SOX2, KLF4, L-MYC and LIN28A; Control 2: Transgenic reprogramming with transgenic SOX2, KLF4, L-MYC and LIN28A, and OCT4 Figure S2b. Control group without addition of dCas9 activators is needed.

(5) dCas9VP192 is more efficiency in iPSC reprogramming (Figure 3g and S2). What is the explanation to this observation?

6. Transcriptional analysis of CRISPRa (effect of EEA). RNA sequencing was conducted to evaluate

the effect of EEA-motif targeting on gene expression at day 4, 8, and 12 after transfection. It should be possible to present the expression of CRISPRa effector and gRNAs from these RNA seq data.

It's a pity that control group with CRISPRa activator alone was not included in this experiment. This group should be served as the basal comparison to define genes activation by OMKSL+KM or by EEA at Day 4.

Figure 4d Day 4, commonly upregulated genes were found. Is there a cluster of down-regulated genes that are EEA-associated based on the same analysis criteria?

When calculating EEA gRNA binding sites, what is the criteria? How many mismatches between guide and target site is allowed?

Legend for Figure 4d needs correction: OMKSL should be OMKSL+KM?

7. Association between NANOG, REX1 and EEA-targeting. Figure 5a. Is there 0 read count for EEA and OMKSL+KM? Based on the ICC data, I would expect to detect certain read count at least for OMKSL+KM group.

Figure 5b. What is defined as control here? VP192 + OMKSL+KM?

Figure 5c. These data are from HEK293 cells. It's a little bit unclear how were the HEK293 cells treated? Are these cells subjected to antibiotic selection post transfection? Also, addition of EEA gRNAs increases NANOG and REX1 expression. Is this effect only limited to the condition that there are NANOG and REX1 targeting gRNAs? Similarly, experimental procedure for Figure S4 should be described more clearly in the legend as well.

General comment: To evaluate EEA associated reprogramming factors, the study compared Day 12 samples in fibroblasts and day 3 (? , 72 hours after transfection if consistently) sample in HEK293. The expression data presented in HEK293 cells reflect EEA CRISPRa-mediated gene activation. However, the gene expression changes at the day 12 reprogrammed fibroblasts is more likely related to enrichment of iPSC forming colonies.

8. Mechanism of EEA-motif targeting. This part of work is very convincing and significant. Figure 6b. Appropriate reference control should be transfected with scrambled gRNA TdT ctrl.

Figure 6e is very confusing and required more detail explanation in figure legend. E.g. what is the transgenic reprogramming? Same as Figure S2? (which transgenic cocktail, as there are two described?)

Is VP192 refers to dCas9-VP192 or VP192 domain only; "no dCas9" referring to no dCas9-VP192?

If would be better if ChIP-seq experiment was conducted to validate that the ATAC peaks with EEA-g1 sites are directly bond by dCas9GFP.

Kr,
Yonglun Luo, Associate Professor
Department of Biomedicine, Aarhus University

Reviewer #2 (Remarks to the Author):

This study investigates the possibility of adopting a fully-based CRISPR/Cas9 cellular

reprogramming technique by targeting the endogenous promoter of canonical reprogramming factors. Building on previous work showing success in the substitution of transgenic Oct4 with a CRISPRa mediated activator (Balboa et al. 2015), the authors engineer a reprogramming methodology exclusively based on the use of a dCas9 protein fused with an activator domain, together with a plasmid containing gRNAs targeting Oct4, Sox2, Klf4, Myc and Lin28a. Moreover, the authors investigate the contribution of simultaneously targeting EEA motifs to the reprogramming efficiency.

Overall, the manuscript presents a novel and exciting observation, that targeting Cas9 itself (in the absence of any transcriptional activation domain) to an EEA motif is as efficient at inducing reprogramming as all the constructs engineered in this study. If this is true, it implies that the binding of Cas9 alone to Alu motifs is sufficient to induce transcription and would represent a major observation, certainly worthy of publishing in Nature Communications. However, the paper does not provide a set of controlled experiments that truly show this is the case or a compelling explanation for why this should be as efficient as their Cas9-activator fusions. For publication, this observation needs to be extensively explored.

Please find some major and minor concerns below.

Major

- 1) The text and the manuscript structure in general are incredibly confusing, with the authors constantly changing between different activator domain-Cas9 fusion proteins (VPH/VP192/VPP300) and starting populations (primary skin fibroblasts/neonatal foreskin fibroblasts). I truly do not understand why there is no consistency in these specific conditions.
- 2) The most interesting observation in the manuscript is that Cas9 itself with EEA can stimulate reprogramming with high efficiency. However, the authors need to show that dCas9 alone, provided that an Alu repeat gRNA is delivered at the same time, has the same efficiency as any other of their suite of fusion proteins in different reprogramming contexts. This needs to take the form of a consistent set of experiments containing relevant controls (+/- EEA, +/- OMKSL gRNA, +/- different Cas9-activation domains). If they are able to induce reprogramming with the EEA gRNA, then they must provide supporting data on the transcriptional activation of relevant pluripotency factors in a time course experiment during reprogramming.
- 3) There seems to be no experiment looking at the correlation between the reprogramming efficiency (AP staining) of different constructs for transcriptional activation and the actual transcriptional activation of reprogramming factors (mRNA levels). Could it be that the better transcriptional activator is actually less efficient at reprogramming? Some attempt to show the connection between immediate early Cas9 mediated activation and reprogramming would make this paper much stronger and also address the fundamental observation apparently reported here, that an activation domain is not required to facilitate Cas9 mediated reprogramming.
- 4) The authors need to show that transfection with EEA gRNA has (or not) an effect on reprogramming efficiency in a traditional reprogramming protocol.

Minor

- 1) Figure 1b-c: mRNA quantification of Oct4 levels after TMP supplementation needs to be provided for all the activator constructs. The text is not supported by adequate evidences: "OCT4 activation in selected HEK293 cells resulted in heterogeneous OCT4 expression upon TMP addition with the p300 core (no quantification provided), therefore TetON-DDCas9VPH was used." Why? Could the authors comment on this decision?
- 2) Again, regarding consistency. Why reprogramming described in Figure 3 now adds the targeting of TP53? Why a line requiring additional gRNAs (KM) has been included and compared to lines targeted only with the OMKSL gRNA plasmid?
- 3) The graph in Figure 2d is very confusing. It seems that VPP300 is the activator domain giving some of the highest activation levels, whereas VPH is the lowest apart from C-Myc expression. Can the authors please explain why they presented this kind of "validation" without properly commenting on the results and decided to keep using the VPH domain?
- 4) Line 189. "We next tested the impact of different dCas9-fused effector domains on the EEA-g1 effect in conventional transgenic reprogramming". Only VP192 is present in the graph. Please

include the rest of the activator domains. Moreover, the Y axis in Figure 6e should show the number of AP+ colonies per million cells, as all the rest of the bar graphs in the manuscript.

5) Figure 5a: To truly understand the relation between NANOG, REX1 and the EEA motifs, the authors should perform reprogramming in cells lacking the expression of the factors, and then see if EEA still have an effect on reprogramming efficiency. Is the observed behavior a result of an additive or an independent effect?

6) Figure 6a: How does the effect of single EEA-gRNA on CRISPR reprogramming compares to the pool of 5 guides used throughout the paper?

Reviewer #3 (Remarks to the Author):

In this work Weltner and colleagues described the use of dCas9 with sgRNAs to target endogenous OCT4, SOX2, KLF4, MYC and LIN28A and generate human iPSCs without transgenic at a low efficient about 0.1% AP+ colonies. EEA-motif (Alu-motif) sgRNA could further enhance human somatic cells reprogramming efficient to ~0.7% (AP+ colonies as they showed in Fig3g). The results are clearly interesting. I have Reviewed this work for another journal and it looks like that they have answered most of my question. Please accept as it is.

Response to the reviewers

We would like to thank the Editor and all Reviewers for their thorough and constructive comments on our manuscript. We have restructured the manuscript to improve the clarity of the text and incorporated the controls requested. The main changes are:

- We have reduced the number of dCas9 activators to two and moved the data dealing with other activators to the supplementary materials to improve clarity.
- We have included additional control inductions for fibroblast reprogramming.
- We have added more data on the characterization of NSC-derived iPSCs.
- We have included control samples for gene activation.

We believe these changes significantly strengthen the message of the study and highlight the relevance of the approach.

We attach a clean version of the revised MS and a version showing text edits. Point-by-point responses to the reviewers' comments are presented below.

====

Reviewer #1:

We would like to thank the Reviewer 1 for the thorough, detailed and constructive revision of the manuscript, which has prompted us to improve its structure, clarity and strengthen the main message.

“Successful reprogramming is typically achieved in only a small portion of cells. In line 64-65 (Fig. 1b), although VPP300 and VPPH fusions achieve a heterogeneous OCT4 activation, the OCT4 signal of positive cells are stronger in those positive cells. For this experiment, firstly, flow cytometry should be conducted to analyze the OCT4 activation in a quantitative manner.”

Reply: In order to increase the clarity of the manuscript we have moved all data concerning the VPP300 transactivator into Supplementary materials (Fig. S4). Accordingly, we have also removed the OCT4 staining pictures from the current version of Figure 1. We performed analysis of global Histone 3 acetylation, as suggested and discussed below, and saw an increase in H3 acetylation bright cells with P300 core activators. It is possible that this observed increase in global histone 3

tail acetylation (Fig.S4 e and f) may contribute to the heterogeneity in OCT4 expression seen in HEK293. To best assess the possible link between the targeted gene activation and the global H3 acetylation, these two should be assessed in temporal fashion in a clonal reporter cell line. We are aiming to study this connection in our future research, but we believe it is not feasible to produce the required cell material in time for this revision. Therefore, we have seen best not to include the HEK293 staining pictures in the revised version.

” Secondly, what is the reason that heterogeneous is a “negative” criteria for choosing VPH rather than VPP300 and VPPH?”

Reply: In order to reliably activate two or more targeted genes in the same cell, we prefer to use activators that predictably activate target genes as homogeneously as possible.

“As commented in Line 61, robust activation of OCT is critical of the success of reprogramming. VPP300 will give the best reprogramming outcome, unless VPP300 creates some unexpected negative effects. P300 CORE has been used in previous study for programmable histone acetylation. However, has the authors checked whether overexpressing P300 CORE will create unspecific histone acetylation. e.g. using anti-histone acetylated antibodies? Control staining is required as supplementary: (1) Un-transfected; (2) Transfected with the fusion protein only (+ and - TMP); (3) Transfected with a scrambled gRNA ((+ and - TMP).”

Reply: We have added analysis of Histone 3 tail acetylation in HEK293 cells transiently transfected with the various activators and guides. This can be found in the Supplementary Figure 4 e and f.

Immunocytochemical staining of Histone 3 tail acetylation shows presence of a subpopulation of cells with bright H3ac staining, indicative of global increase in nuclear H3 tail acetylation, in P300 core containing activators and particularly with VP192-P300 fusion. The amount of H3 tail acetylation bright cells was quantified by FACS in dCas9 activator and TdT control gRNA transfected HEK293. An increase in H3 acetylation bright cells can be seen as an elongated tail in the FACS histograms (Sup. Fig. 4f). This increase in global H3 acetylation may end up affecting the final gene activation and reprogramming efficiency, as histone deacetylase inhibitors used in high concentrations can decrease their effect on reprogramming (Mali et al. 2010 PMID: 20201064).

“2. All CRISPRa effector proteins contain a GFP, does this create problem for the immunostaining using green channel? “

Reply: The GFP fluorescence from the vectors is generally not an issue. The GFP fluorescence from the dCas9 activator plasmids we have used peaks at day 2 after electroporation and is weak after

that. All the iPSC line analyzed have been GFP negative. All transiently transfected cells have been stained with red secondary antibody, except the OCT4 staining in the Figure 5 inductions. In the OCT4 staining in Figure 5 the main gene analyzed for its activation is NANOG, which is stained in red, and OCT4 serves as a marker for the colony formation. The staining pattern and intensity of OCT4 is different from the dCas9 activator plasmids as the GFP in the vectors is fused with a 2A sequence and is not primarily localized to the nucleus.

“3. CRISPRa reprogramming of NPC: Although the system used is DOX and TMP inducible, it is critical to show that: (1) The CRISPRa-derived iPSCs are GFP negative.”

Reply: Picture of the absence of GFP fluorescence in the NSC derived iPSCs has been added in the Supplementary Figure 1b.

The CRISPRa derived iPSCs are generally not morphologically stable if the dCas9 activator is actively expressed or leaky in the cells. A picture of the induction of dCas9VPH by doxycycline and trimethoprim addition in OCT4 gRNA containing NSC derived iPSCs has been added to the Supplementary Figure 1c.

“(2) It’s important to show that these are real CRISPRa-derived iPSCs. E.g. by PCR validating the presence of CRISPRa effector gene in iPSC genomic DNA.”

Reply: Picture of genomic DNA PCR for the Cas9 amplicon has been added in the Supplementary Figure 1a.

“(3) The iPSC-derived NPCs used for reprogramming are generated by Sendai reprogramming, although this method has very low risk of transgene integration, I cannot find the information about transgene integration testing in their previous paper (Stem Cell Research (2015) 15, 254–262).”

Reply: The iPSC used for NSC generation have been HEL24.3 and HEL46.11 lines. Derivation of HEL24.3 has been described in Stem Cell Research cell line paper (10.1016/j.scr.2015.05.012) and HEL46.11 RT-PCR for the Sendai virus replicon is included in the Supplementary Figure 2e.

“(4) What about the efficacy of CRISPRa-mediated OCT4 activation as compared to expressing transgene OCT4 in NPCs? Reprogramming efficiency between overexpressing transgene OCT4 VS. CRISPRa mediated activation of OCT4? OCT4 Gene expression level by qPCR?”

Reply: Transgenic OCT4 expression was included in some of the NSC inductions we performed, but these cells did not give consistent reprogramming results. Sometimes transgenic OCT4 expression has failed to reprogram NSCs, whereas other times it produces small tight alkaline phosphatase positive colonies. A picture of transgenic OCT4 reprogrammed alkaline phosphatase stained plate has been added to the Supplementary Figure 1d.

Targeted gene activation has now been included in more detail in figure 2d and analysed in HEK293 and fibroblasts.

“(5) As I can see from the M&M, there are two iPSC-derived NPC lines used for testing. Do both lines give equally good reprogramming?”

Reply: One induction experiment has been made with HEL46.11 derived NSC and the rest have been made with HEL24.3 derived NSCs. This is not sufficient to tell if there is a cell line specific difference in reprogramming efficiency.

“(6) What about none iPSC-derived NPCs?”

Reply: We have tested CRISPRa mediated *OCT4* activation in primary human NSC, but these cells differentiated rapidly after electroporation and we were not able to reprogram them even though *OCT4* activation could be seen by immunostaining. A picture of primary NSC with CRISPRa mediated *OCT4* targeting is included below.

--

“(7) It’s important to have results from two extra controls to conclude that “Line 69, CRISPRa mediated activation of OCT4 alone was sufficient to ...”: (a) Expression CRISPRa effector only; (b) CRISPRa with scrambled gRNAs.”

Reply: The NSC reprogramming experiments have originally been made with controls for cells that have not been treated with doxycycline and trimethoprim and cells that have been targeted by the EEA-motif gRNAs. Neither of these conditions produce alkaline phosphatase positive colonies. This demonstrates that the cells do not convert back to iPSCs by themselves nor by the activator alone. These conditions were initially excluded from the graph to simplify the figure, but these results have now been added to the Figure 1 e graph.

“4. EEA targeting. Again, Fig 1h. GFP transfection is the not the appropriate control to conclude that EEA-motif targeting could be useful for improving reprogramming efficiency. None-targeting scrambled gRNAs should be used to compare with (similar to what Figure 6a).”

Reply: We have repeated the transgenic fibroblast reprogramming with dCas9VP192 and dCas9VPH mediated targeting of EEA-motif using non-genomic TdTomato sequence targeting gRNA as a control. The effect of EEA-motif targeting in transgenic reprogramming in these experiments was much weaker than the previous experiment and not statistically significant. This would indicate that the previously used pCXLE-GFP control may have a negative effect on the overall reprogramming efficiency. This may be due to differences in the plasmid backbone affecting the behavior of the vectors in reprogramming. We have replaced the experiment with a proper control induction. The data are presented in the Supplementary Figure 3a.

The effect of the EEA-motif targeting appears to depend on the overall reprogramming efficiency. This is a likely contributing factor to the EEA-motif targeting effect on CRISPRa reprogramming, as the basal reprogramming efficiency with just pluripotency factor targeting gRNAs is very low. We have thus also included EEA-motif targeting in transgenic reprogramming using OCT4, SOX2, LIN28 and L-MYC, in the absence of transgenic KLF4, as this prevents reprogramming in the control conditions. EEA-motif targeting with dCas9 activators in the absence of transgenic KLF4 does produce colonies (Fig. 6 g and Fig. S7 c).

“Fig 1e. EEA gRNAs used are truncated gRNAs, please state that in the article.”

Reply: The length of the guide RNAs has been added to the text.

“Also, it would be better to demonstrate that CRISPRa binds to the EEA consensus sequences. E.g. by Chip-seq.”

Reply: The binding of the EEA-gRNAs to the motif can be seen in the reporter targeting experiments in Figure 6 b and c.

We agree that CHIP-seq of the dCas9 targeting the EEA consensus sequence would be optimal to determine directly where it binds in the genome. However, in order to yield useful data in the correct context, the effect of the EEA-motif targeting will need to be assessed in the reprogramming process. This will require the isolation of high amounts of cells in undergoing reprogramming and is not yet feasible using the current CRISPRa reprogramming approach due to the low efficiency of the current system.

“Fig. 1. g-h. The experiment has been tested in 5 independent inductions. Are these from the same NPC line?”

Reply: The NSC reprogramming experiment (currently Fig 1e) has been made with one induction of HEL46.11 derived NSC and the rest with HEL24.3 derived NSC at different passages and from different differentiation batches.

“Fig. 1h. There were 8.7-fold increase in reprogramming efficiency (transgenic reprogramming + EEA CRISPRa), VPH used. How is this experiment compared to Figure 6e? (VP192).”

Reply: We performed the EEA-motif targeting experiments again using TdT-gRNA as a control as described previously. The revised EEA-motif targeting inductions with transgenic pluripotency factors and the TdT guide control are closer to the efficiencies seen in Figure 6e. However, as the plasmid composition seems to affect the reprogramming efficiency quite drastically, we have also performed the Figure 6 transgenic reprogramming with weaker reprogramming conditions by removing transgenic KLF4 from the mixture. These new data have been included as the Figure 6g. In the absence of KLF4 we see AP+ iPSC-like colonies only in the presence of dCas9 activators. This further indicates that the effect of the EEA-motif targeting is dependent on the activatory function of the dCas9 effector targeted to the motif. Moreover, this suggests that the EEA-motif targeting may complement for the poor activation of the CRISPRa targeted *KLF4*, as seen in fibroblasts in Figure 2d. CRISPRa reprogramming may therefore be less dependent on high KLF4 overexpression when EEA-gRNAs are included

“5. Fibroblasts reprogramming: (1) what is the basal expression level of these five genes in HEK293 cells? It would be difficult to activate genes that are already highly expressed. For example, MYC should be highly expressed.”

Reply: We have included mRNA quantification by qRT-PCR for activated genes for both HEK293 and HFF relative to H9 hESC expression levels. MYC expression levels are high in HEK293 cells but we can detect approximately 20% increase in its expression in dCas9VPH and OMKSL guide plasmid transfected cells. We do not see significant upregulation at the population level in electroporated HFFs for KLF4, MYC and LIN28.

“(2) Line 83-84, based on Figure 2d, it’s a little bit hard to agree that all four effectors showed comparable gene activation potential. VPP300 apparently outcompete all the other three. The MYC gene is highly expressed in HEK293 cells, (TPM value equal to approximately 386 according to published RNA-seq data), the activation data from this gene in HEK293 cells cannot contribute to the conclusion.”

Reply: The activators showed consistent and statistically significant difference in activation efficiency in both day 1 and day 3 time points only for *OCT4* activation with VPP300. We have added a more comprehensive discussion about the different activators in the Supplementary note and the accompanying Supplementary Figure 4.

“(3) Comments regarding: Reprogramming of fibroblasts into iPSC by CRISPRa. Line 85-97, Figure 3 and Supplementary Figure 1. (3.1). Figure S3c should not appear and cited before Figure S1 Line 90. Re-order the supplementary figures.”

Reply: The Figure numbering has been changed in the revised manuscript version.

“(3.2) Figure S1c, Karyotyping lack clone HEL136.2.”

Reply: We have not been able to receive karyotyping results of this particular cell line due to technical difficulties with its culture maintenance and expansion. As we cannot confirm the karyotype of the line we have excluded it from the manuscript and re-analyzed the expression data without the HEL136 line.

“(3.3). the reprogramming is based on plasmid electroporation, was antibiotic selection used to enriched transfected cells?”

Reply: Antibiotic selection has not been used in the episomal plasmid reprogramming. Since the peak of the puromycin resistance marker expression occurs during the post-electroporation recovery period (2-4 days), selection with puromycin results in excessive cell toxicity and impaired reprogramming. On the other hand, in transposon-based reprogramming, the cells can be selected after post-electroporation recovery and the CRISPRa can be induced after selection.

“(3.4). Figure S1d, EBNA PCR showed that there is integration in some iPSC clones. Plasmid integration is randomly break and inserted. Could it be possible that the EBNA PCR fails to detect the integration? What about PCR of dCas9?”

Reply: We have added the Cas9 genomic DNA PCR to the Supplementary Figure 2. We have also checked the RNA sequencing data of the CRISPRa iPSCs and found no reads from the dCas9 constructs. In order for the CRISPRa system to work it needs both dCas9 and guide components, therefore the absence of Cas9 demonstrates the independence of the established clones' pluripotent state from the CRISPRa mediated gene activation.

“(3.5). Figure 3d-e, legends for PCA do not corresponding to the plot.”

Reply: The PCA plot legends have been corrected.

“The PCA analysis is based on “fluctuated genes, 268” and “165 significantly DEG, for the heat map plot”. If these genes are preselected based on comparison between iPSCs and fibroblasts, there is no doubt that the samples will be clustered separately. What about PCA analysis based on all genes?”

Reply: The fluctuated genes are also known as variable genes e.g. in Brennecke *et al.* [PMID: 24056876], therefore the 268 genes for PCA (before exclusion of HEL136) were not preselected by a comparison of cell types. We have also performed PCA of the samples based on all 5312 detected genes. This separates the cells similarly to the smaller set primarily by pluripotent cell state in the PC1 axis and primarily by the cell line in the PC2 axis. A picture of the PCA using all genes has been included below for revision.

--

“(3.6) Figure 3g. What is used as control (ctrl)? dCas9 activator with OMKSL gRNAs?”

Reply: These inductions have been made with following plasmid compositions:

ctrl: pCXLE-dCas9 activator (2 µg) + GG-EBNA-OMKSL-PP (2 µg) + GG-EBNA-KM-PP (2 µg)

EEA: pCXLE-dCas9 activator (1.5 µg) + GG-EBNA-OMKSL-PP (1.5 µg) + GG-EBNA-KM-PP (1.5 µg) + GG-EBNA-EEA-5guides-PP (1.5 µg)

“(4) Expressing dCas9VPH alone and transgenic factors decreased reprogramming efficiency. Figure S2 is normalized and compared to VP192. What is the reprogramming efficiency when comparing to transgenic factors only? Figure S2a. Controls required: Control 1: Transgenic reprogramming with transgenic SOX2, KLF4, L-MYC and LIN28A Control 2: Transgenic reprogramming with transgenic SOX2, KLF4, L-MYC and LIN28A, and OCT4 Figure S2b. Control group without addition of dCas9 activators is needed.”

Reply: These control inductions have been added to the Supplementary Figure 3d. The control inductions have not been added to the same graphs (b and c) as the other inductions as they are from separate experiments. The Inductions are presented as relative to transgenic OCT4, SOX2, KLF4, LIN28, MYC and dCas9VP192 which allows comparison to other graphs (b and c).

“(5) dCas9VP192 is more efficiency in iPSC reprogramming (Figure 3g and S2). What is the explanation to this observation?”

Reply: This is an excellent question. The higher performance of the weaker dCas9VP192 is counterintuitive to what would be expected. We do not know exactly what the causative factors are behind this effect. We observe a decrease in reprogramming efficiency in the presence of dCas9VPH activator even in the absence of guide RNAs, which would indicate that the VPH domain may have a negative effect on reprogramming efficiency. This could be explained by interference of the fusion transactivator domain with their associated signaling pathways (e.g. p65 and HSF1). VPP300 and VPPH show reduced reprogramming efficiency when targeting *OCT4* or other genes (Fig. S3b), but not in fully transgenic transcription factor mediated reprogramming (Fig. S3c). These effects may be partially explained by i) guide RNA dependent off-target dCas9 binding and acetylation, ii) negative on-target effects that may affect *OCT4* expression (e.g. sterical impediment effects or interfering acetylation), iii) excessive *OCT4* expression resulting in imbalanced reprogramming transcription factor stoichiometry or iv) the size of the plasmids affecting their delivery efficiency and replicative maintenance. Additionally, we do not know the effect of the different activation domain fusions on each other and whether they interfere sterically with each other.

The interaction of the activator domains with the reprogramming process is a complicated but interesting question and worthy of further investigation. However, we believe this goes beyond the scope of this study.

“6. Transcriptional analysis of CRISPRa (effect of EEA). RNA sequencing was conducted to evaluate the effect of EEA-motif targeting on gene expression at day 4, 8, and 12 after transfection. It should be possible to present the expression of CRISPRa effector and gRNAs from these RNA seq data.”

Reply: There is higher dCas9 expression in day 4 samples that decreases in the later samples. We found no differences between the different sample conditions (EEA, OMKSL + KM, EEA + OSKML + KM). Guide RNA expression may not be reliably detected due to Pol III driven expression of their transcripts and poly A priming used in STRT-seq sample preparation. We have included a picture of the dCas9 normalized read counts at different time points of reprogramming for revision.

dCas9 reads in CRISPRa reprogramming RNA-seq

--

“It’s a pity that control group with CRISPRa activator alone was not included in this experiment. This group should be served as the basal comparison to define genes activation by OMKSL+KM or by EEA at Day 4.”

Reply: Yes, we agree. The exclusion of electroporated negative control was due to practical limitations in the sequencing library size.

“Figure 4d Day 4, commonly upregulated genes were found. Is there a cluster of down-regulated genes that are EEA-associated based on the same analysis criteria?”

Reply: As stated in the previous question, the comparison of the samples between different time points and the HFF control is not reliable due to changes in gene expression caused by the electroporation. We have changed the text to reflect this and renamed the gene sets accordingly to ‘higher expressed’ and ‘lower expressed’ rather than ‘upregulated’ and ‘downregulated’. The genes showing lower expression in day 4 samples than in control HFF in figure 4b heatmap (group 4) are associated with stress fiber formation and extracellular matrix organization.

“When calculating EEA gRNA binding sites, what is the criteria? How many mismatches between guide and target site is allowed?”

Reply: EEA-g1 binding sites have been calculated with perfect matches to the EEA-g1 sequence, due to strongest effect of this guide on reprogramming efficiency (figure 6a) and perfect matching being important for high affinity targeting.

“Legend for Figure 4d needs correction: OMKSL should be OMKSL+KM?”

Reply: The legend has been corrected.

--

“7. Association between NANOG, REX1 and EEA-targeting. Figure 5a. Is there 0 read count for EEA and OMKSL+KM? Based on the ICC data, I would expect to detect certain read count at least for OMKSL+KM group.”

Reply: Yes, there is zero read count at day 4 for *LIN28A*. This indeed contrasts with the *LIN28A* expression activation detected by ICC in a minor population of the cells at day 4. This suggest that the cDNA library sequencing depth was not sufficient to reliably detect the activation of all the targeted factors at day 4, probably due to the much more abundant fibroblast background transcripts. This is the reason why we used a two-step approach to validate the EEA-motif targeting gene candidates in improving reprogramming, as is further discussed below.

“Figure 5b. What is defined as control here? VP192 + OMKSL+KM?”

Reply: The control condition in these inductions is dCas9VP192 and OSK2M2L1 plasmids as in Supplementary Figure 3a. This has been added to the figure labels.

“Figure 5c. These data are from HEK293 cells. It’s a little bit unclear how were the HEK293 cells treated? Are these cells subjected to antibiotic selection post transfection?”

Reply: These cells were transiently transfected. This information has been now added to the text. The cells were not treated with antibiotics.

“Also, addition of EEA gRNAs increases NANOG and REX1 expression. Is this effect only limited to the condition that there are NANOG and REX1 targeting gRNAs?”

Reply: The activation of *REX1* is not limited to *REX1* gRNA containing conditions. *REX1* activation can also be seen in the presence of *NANOG* gRNAs + EEA gRNAs (Figure S6c). The presence of *NANOG* gRNAs increased the expression of *REX1* over 2-fold compared to non-treated cells (Figure S6c). Additional EEA gRNAs enhanced activation efficiency (Figure S6c), possibly due to an EEA-guide 1 site between the *REX1* gRNAs sites.

The activation of *NANOG* is not limited to *NANOG* and EEA-gRNA containing condition. Although *NANOG* is not activated by EEA-motif targeting directly (Figure S6c), more rapid *NANOG* activation can be seen by immunocytochemistry in OSK2M2L1 gRNA-mediated CRISPRa reprogramming without *NANOG* gRNAs (Figure 5d).

“Similarly, experimental procedure for Figure S4 should be described more clearly in the legend as well.”

Reply: This has been added to the figure legend, previous Figure S4 is current Figure S6.

“General comment: To evaluate EEA associated reprogramming factors, the study compared Day 12 samples in fibroblasts and day 3 (? , 72 hours after transfection if consistently) sample in HEK293. The expression data presented in HEK293 cells reflect EEA CRISPRa-mediated gene activation. However, the gene expression changes at the day 12 reprogrammed fibroblasts is more likely related to enrichment of iPSC forming colonies.”

Reply: The gene expression changes at day 12 are most likely associated with the emergence of pluripotent stem cell colonies and the expansion of the PSCs in the sample cell population. This is the reason we took the two-step approach to test for genes possibly affected by EEA-motif targeting.

1. Assuming that the EEA-gRNA affected genes stay transcriptionally activated during the reprogramming process, these genes should remain activated in the expanded pluripotent stem cell population at day 12. These genes are initially activated in such small sub-population that we cannot detect them reliably in the pool of reprogramming cells. As the reprogrammed cell population is expanding, these genes become detectable. For these genes to be EEA-motif targeting candidates for mediating the improved reprogramming, these genes should promote reprogramming in the absence of EEA-gRNAs, as *NANOG* and *REX1* did (Figure 5b). This approach will miss EEA-motif targeting candidates which are only transiently affected.

2. If the candidate genes from the first step are to be EEA-motif targeting effectors, their activation should also be improved by EEA-gRNAs. This was tested by directed activation of *NANOG* and *REX1* in Figure 5c.

More thorough analysis of the EEA-motif targeted genes in CRISPRa reprogramming will require improved methods for isolating cells undergoing reprogramming and more efficient CRISPRa reprogramming, particularly in the absence of the EEA-gRNAs. Although this is not currently feasible, we are working on improving these methods to enable this kind of analysis in the future.

“8. Mechanism of EEA-motif targeting. This part of work is very convincing and significant. Figure 6b. Appropriate reference control should be transfected with scrambled gRNA TdT ctrl.”

Reply: We could not use the previously used TdTomato guide as a control in this experiment as the ORF of mCherry contains a binding site for the TdT-gRNA. We used pMXs-DD-GFP as a filler DNA to keep the DNA amounts similar in the transfection. We do not think that the used GFP plasmid should have a negative effect, as it is not episomally maintained in the cells. We have added pictures of the mCherry reporter expression in the Supplementary Figure 7b.

“Figure 6e is very confusing and required more detail explanation in figure legend. E.g. what is the transgenic reprogramming? Same as Figure S2? (which transgenic cocktail, as there are two described?). Is VP192 refers to dCas9-VP192 or VP192 domain only; “no dCas9” referring to no dCas9-VP192?”

Reply: We apologize for the confusing charts and labeling of reprogramming conditions. We have now added under each chart the details on the reprogramming conditions used in each experiment.

“If would be better if ChIP-seq experiment was conducted to validate that the ATAC peaks with EEA-g1 sites are directly bond by dCas9GFP.”

Reply: This would be optimal, but as the EEA-motif targeting is likely to be affected by the reprogramming process, the most relevant information would be gained from ChIP-seq samples of the reprogramming cells. This is something that we are aiming to do in order to understand better the functional mechanism of EEA motif targeting. However, this is not yet possible with our current technology as it will require significant improvement of the reprogramming efficiency and the cell selection methods due to the low efficiency of the current CRISPRa reprogramming system.

====

Reviewer #2:

We thank the Reviewer 2 for the observations raised that have helped us improve the clarity of the manuscript.

“Overall, the manuscript presents a novel and exciting observation, that targeting Cas9 itself (in the absence of any transcriptional activation domain) to an EEA motif is as efficient at inducing reprogramming as all the constructs engineered in this study.”

Reply: The dCas9 mediated targeting of the EEA-motif does not by itself reprogram cells. This requires presence of either transgenic reprogramming factors or activation of endogenous reprogramming factors by CRISPRa. We admit that this important point might have not been presented with enough clarity in the manuscript. We have improved the clarity of the presentation throughout the manuscript, for example, by including additional details on the reprogramming conditions in each experiment and describing more thoroughly the reprogramming factors in the text.

--

“1) The text and the manuscript structure in general are incredibly confusing, with the authors constantly changing between different activator domain-Cas9 fusion proteins (VPH/VP192/VPP300) and starting populations (primary skin fibroblasts/neonatal foreskin fibroblasts). I truly do not understand why there is no consistency in these specific conditions.”

Reply: We have improved the clarity by reducing the dCas9 activator in the main text to only two (dCas9VP192 and dCas9VPH) and including the P300 core containing activators (dCas9VPP300 and dCas9VPPH) only in the supplementary materials.

As the P300 core domain containing activators were not used in the reprogramming experiments, for other than testing their efficiency for colony formation, they have been excluded from the main text. We have included the P300 core factors in a supplementary note for readers who may be interested in the dCas9 activator development. These activators may prove to be more useful, for example, in different vector or reprogramming contexts.

--

“2) The most interesting observation in the manuscript is that Cas9 itself with EEA can stimulate reprogramming with high efficiency.”

Reply: dCas9 targeting to the EEA-motif improves the reprogramming outcome with transgenic reprogramming by OCT4, SOX2, KLF4, LIN28A, and L-MYC. This was not presented clearly enough in the previous version and has now been changed to improve clarity (Fig. 6e). The EEA-motif targeting still needs the presence of either pluripotency factor targeting guides or some set of transgenic pluripotency factors for reprogramming to happen. The EEA-motif targeting effect in improving reprogramming appears to be more prominent in low efficiency reprogramming conditions. Inclusion of EEA-motif targeting gRNAs is crucial for efficient CRISPRa reprogramming, particularly with primary adult fibroblasts. Therefore, the EEA-motif targeting seems to lower the barrier for cell type conversions in pluripotent reprogramming, but it is not sufficient to complete the process by itself.

The effect of the EEA-motif targeting it does not appear to be dependent on the dCas9 fusion domain if all transgenic reprogramming factors are used (OCT4, SOX2, KLF4, LIN28 and L-MYC) probably due to already relatively high basal reprogramming efficiency. The reprogramming of fibroblasts with reduced set of transgenic factors, without KLF4 (OCT4, SOX2, LIN28 and L-MYC), requires an activator domain and does not produce colonies by default (Figure 6g and Supplementary Figure 7c).

--

“3) There seems to be no experiment looking at the correlation between the reprogramming efficiency (AP staining) of different constructs for transcriptional activation and the actual transcriptional activation of reprogramming factors (mRNA levels).”

Reply: Transcriptional activation of the targeted pluripotency factors with different activators in HEK293 are included in the Supplementary Figure 4. Transcriptional activation of the pluripotent reprogramming factors with dCas9VPH in HEK293 and fibroblasts are presented in the Figure 2d. Reprogramming efficiency with dCas9VP192 and dCas9VPH are presented in the Figure 3g and reprogramming efficiency with dCas9VPP300 and dCas9VPPH are presented in the Supplementary Figure 4d.

There is no clear correlation of the activation efficiency of the factors and the reprogramming efficiency. It is possible that some of the activator domain constructs have negative effect on the reprogramming efficiency by themselves. This appears to be the case with dCas9VPH, which reduces reprogramming efficiency when added to otherwise transgenic factor (OCT4, SOX2, KLF4, LIN28, L-MYC) mediated reprogramming (Supplementary Figure 3c).

Additionally, we have analyzed global Histone 3 acetylation in P300 core domain containing dCas9 effector transfected HEK293 cells for the revision. The transfection of HEK293 cells with dCas9VPP300 or dCas9VPPH with control TdT-gRNA results in an increase in H3 acetylation bright cell numbers (Supplementary Figure 4e and f). This increase in global H3 acetylation may end up affecting the final gene activation and reprogramming efficiency as histone deacetylase inhibitors used in high concentrations can decrease their effect on reprogramming (Mali et al. 2010 PMID: 20201064).

--

“4) The authors need to show that transfection with EEA gRNA has (or not) an effect on reprogramming efficiency in a traditional reprogramming protocol.”

Reply: The reprogramming of fibroblasts with transgenic OCT4, SOX2, KLF4, LIN28A and L-MYC and the EEA-gRNA in the absence of dCas9 effector is presented in the Figure 6e. This chart was probably too unclear previously and we have improved the clarity of the figure by adding all the included reprogramming factors under the chart and in the text.

“1) Figure 1b-c: mRNA quantification of Oct4 levels after TMP supplementation needs to be provided for all the activator constructs. The text is not supported by adequate evidences: “OCT4 activation in selected HEK293 cells resulted in heterogeneous OCT4 expression upon TMP addition with the p300 core (no quantification provided), therefore TetON-DDCas9VPH was used.” Why? Could the authors comment on this decision?”

Reply: We have changed this figure upon re-structuring of the manuscript to improve its clarity. Thus, we have also removed these data from the manuscript. We performed analysis of global Histone 3 acetylation, as a part of the revision, and saw an increase in H3 acetylation bright cells with P300 core activators. It is possible that the increase seen in global histone 3 tail acetylation (Fig.S4 e and f) may contribute to the heterogeneity in OCT4 expression seen in HEK293. To best assess the possible link between the targeted gene activation and the global H3 acetylation, these two should be assessed in temporal fashion in a clonal reporter cell line. We are aiming to study this connection in our future research, but we think it is not feasible to produce the required cell material in time for this revision. Therefore, we have seen best not to include the HEK293 staining pictures in the revised version.

The reason for trying to avoid heterogeneous activation of targeted genes in treated cell populations is to promote more homogenous and simultaneous activation of different genes in individual cells when more than one gene needs to be activated. Not all the cells expressing the activators are activating all the targeted genes. In order to reliably activate two or more targeted genes in the

same cell, we aimed at using activators that predictably activate target genes as homogenously as possible.

” 2) Again, regarding consistency. Why reprogramming described in Figure 3 now adds the targeting of TP53? Why a line requiring additional gRNAs (KM) has been included and compared to lines targeted only with the OMKSL gRNA plasmid?”

Reply: TP53 targeting shRNA is included in the plasmid reprogramming protocol to improve the cell survival in the reprogramming procedure after the electroporation. This is commonly used in episomal plasmid based reprogramming methods (e.g. Okita *et al.*, Nature Methods, 2011). The TP53 shRNA is not required for CRISPRa reprogramming *per se*, as shown in the transposon-mediated reprogramming experiments presented in the Supplementary Figure 8, which uses vectors that do not include the TP53 shRNA transcription cassette. Additional KLF4 and MYC guide RNAs improve the reprogramming efficiency probably due to insufficient activation of the targeted genes by single guides. We have added a graph in Figure 2d quantifying the activation efficiency of the pluripotent reprogramming factors in HEK293 and fibroblasts. This graph shows poor population level activation of *KLF4*, *LIN28* and *MYC* in human fibroblasts using only the OMKSL gRNA plasmid supporting the use of additional guides targeting *KLF4* and *MYC*.

“3) The graph in Figure 2d is very confusing. It seems that VPP300 is the activator domain giving some of the highest activation levels, whereas VPH is the lowest apart from C-Myc expression. Can the authors please explain why they presented this kind of “validation” without properly commenting on the results and decided to keep using the VPH domain?”

Reply: P300 core activators and further discussion of the different activators has been moved to the Supplementary note and Supplementary Figure 4. We hope that this clarifies the main messages of the manuscript.

P300 core containing activators, particularly VPP300, improve activation of *OCT4* in HEK293 at both analyzed time points (day 1 and day 3) (Fig. S4c). With other genes the effect is not clear. Long term expression of these constructs in cells may have negative effects on the cells as we see *KLF4*, *MYC* and *SOX2* expression drop in P300 core containing conditions between days 1 and 3 (Fig. S4c). The pluripotent reprogramming process will require persistent expression of the activators until full pluripotent gene expression programs have been properly activated.

We have added analysis of Histone 3 tail acetylation in HEK293 cells transiently transfected with the various activators and guides. This can be found in the Supplementary Figure 4 e and f. Immunocytochemical staining of Histone 3 tail acetylation shows presence of a subpopulation of cells with bright H3 acetylation staining, indicative of global increase in nuclear H3 tail acetylation, in P300 core containing activators and particularly with VP192-P300 fusion. The amount of H3 tail acetylation bright cells was quantified by FACS in dCas9 activator and TdT control gRNA transfected

HEK293. An increase in H3ac bright cells can be seen as an elongated tail in the FACS histograms (Sup. Fig. 4f). This increase in global H3 acetylation may end up affecting the final gene activation and reprogramming efficiency as high amounts of histone deacetylase inhibitors can decrease their reprogramming promoting effect. The possible link between global H3 tail acetylation and defective gene activation would be best studied in clonal human reporter cells, which are not currently available.

--

“4) Line 189. “We next tested the impact of different dCas9-fused effector domains on the EEA-g1 effect in conventional transgenic reprogramming”. Only VP192 is present in the graph. Please include the rest of the activator domains.”

Reply: As the P300 core containing activator are not anymore included in the main text we have not included them in further induction experiments. We have re-done the dCas9 activator domain fusion experiment with suboptimal transgenic reprogramming in the absence of KLF4, containing OCT4, SOX2, LIN28A and L-MYC, as this gives better contrast than full OCT4, SOX2, KLF4, LIN28, L-MYC reprogramming used in Figure 6e. This experiment has been performed with dCas9VP192, dCas9VPH and dCas9GFP and included in Figure 6g. The reprogramming promoting effect of the EEA-motif targeting in the absence of KLF4 is dependent on the activation domain fused to the dCas9 effector as only EEA-g1 containing dCas9VP192 and dCas9VPH conditions show any colony formation (Figure 6g and Supplementary Figure 7c).

--

“Moreover, the Y axis in Figure 6e should show the number of AP+ colonies per million cells, as all the rest of the bar graphs in the manuscript.”

Reply: This has been changed.

--

“5) Figure 5a: To truly understand the relation between NANOG, REX1 and the EEA motifs, the authors should perform reprogramming in cells lacking the expression of the factors, and then see if EEA still have an effect on reprogramming efficiency. Is the observed behavior a result of an additive or an independent effect?”

Reply: The reprogramming of human cells in the absence of NANOG or REX1 is very unlikely to work as both NANOG and REX1 are important for human pluripotent stem cell maintenance. These factors are unlikely to be the sole targets of the EEA-motif targeting as the motif is very common in the human genome (over 360 000 sites for EEA-g1 alone). Additionally, the choice of genes affected by EEA-motif targeting most likely depends on additional factors present in the cells, such as simultaneous promoter targeting guides or reprogramming transcription factors. This is exemplified by the results on the EEA-motif targeting effect on *NANOG* activation. EEA-motif targeting guides promote *NANOG* activation when dCas9 is simultaneously targeted to the *NANOG* promoter (Figure 5c). *NANOG* activation is also promoted by EEA-motif targeting in the presence of pluripotent reprogramming transcription factors (Figure 5d). EEA-motif targeting by itself does not result in *NANOG* activation (Supplementary Figure 6c). Identification of additional targets will require higher efficiency CRISPRa reprogramming methods and purification of reprogramming cell populations.

--

“6) Figure 6a: How does the effect of single EEA-gRNA on CRISPR reprogramming compares to the pool of 5 guides used throughout the paper?”

Reply: The comparison of EEA-g1 and a pool of 5 guides on CRISPRa reprogramming with dCas9VP192 and OSK2M2L1 gRNA plasmid has been included in the Supplementary Figure 7a. These EEA-motif targeting guide plasmids appear to perform with equal efficiency in HFF reprogramming.

====

Reviewer #3:

We would like to thank the Reviewer 3 for the positive comments.

REVIEWERS' COMMENTS:

Reviewer #1 (Remarks to the Author):

The authors have really well addressed all the questions in their revision. And the current revised manuscript is excellent and well improved, and should be published as it is. This study further supports and expands the CRISPR tool in gene activation and reprogramming.

Kr,
Yonglun Luo

Reviewer #2 (Remarks to the Author):

Following revision, the authors have successfully restructured the manuscript, dramatically improving the general level of clarity. Moreover, they satisfactorily answered all my initial concerns. The message of the study has been indeed strengthened, and for this reason I now feel confident in endorsing the publication of this paper in Nature Communications.